# Virtual water trade and water footprint of agricultural goods: the 1961-2016 CWASI database

Stefania Tamea[1], Marta Tuninetti[1], Irene Soligno[1], Francesco Laio[1]

[1]Department of Environment, Land and Infrastructure Engineering, Politecnico di Torino, Torino, 10129, Italy

5 *Correspondence to*: Stefania Tamea (stefania.tamea@polito.it)

**Abstract.** To support national and global assessments of water use in agriculture, we build a comprehensive database of country-specific water footprint and virtual water trade (VWT) data for 357 agricultural goods. The water footprint, indicating the water needed for the production of a good including rainwater and water from surface- and ground- water bodies, is expressed as a volume per unit weight of the good (or unit water footprint, uWF) and is here estimated at the country scale for every year in the period 1961-2016. The uWF is also differentiated, where possible, between production and supply, referring to local production and to a weighted mean of local production and import, respectively. The VWT data, representing the amount of water needed for the production of a good and virtually exchanged with the international trade, are provided for each commodity as bilateral trade matrices, between origin and destination countries, for every year in the period 1986-2016. The database, developed within the CWASI project, improves upon earlier datasets because it takes into account the annual variability of the uWF of crops, it accounts for both produced and imported goods in the definition of the uWF and it traces goods across the international trade up to the origin of goods' production. The CWASI database is available on the Zenodo repository at https://doi.org/10.5281/zenodo.3987468 (Tamea et al., 2020) and welcomes contributions and improvements from the research community to enable analyses specifically accounting for the temporal evolution of the uWF.

## 1 Introduction

There has been a booming interest in the concept of Water Footprint (WF) since its introduction about 15 years ago (Hoekstra & Chapagain, 2007, 2008). The water footprint offers a common approach, language and method to a wide range of analyses and multidisciplinary studies, and it is appreciated for its capability to convey environmental messages to the public. The WF identifies the freshwater needed for the production of goods along the full supply chain, separating rainfall and water from surface/ground-water bodies. The WF assessment provides a quantitative framework to analyse the volume of water embedded in agricultural goods and the efficiency of water use, when the metric is computed per unit weight of the good (hereafter referred to as the unit water footprint, or uWF). The term *unit water*

*footprint* is here introduced to unify the current terminology which includes "water footprint", used indifferently for volumes and for volumes per unit weight, "crop water footprint" which excludes livestock products, or "virtual water content" mainly used within the context of trade (see for e.g., Hoekstra et al., 2011; Konar et al., 2011; Dalin et al., 2012; Tuninetti et al., 2015). Also the concept of virtual water, originally proposed by J. A. Allan (1998) and from which the WF originated, has been growing in popularity among both the scientific community and the general public. Virtual water is the volume of water needed to produce a certain good that is virtually traded as a factor of production when the good is exchanged among countries. Such virtual flow defines the international virtual water trade (VWT) and represents a metric that is suitable to analyse environmental aspects related to the global trade of agricultural goods, to the water management and to the agricultural policy.

Assessment of WF and VWT requires a relatively large amount of data, including production and trade data (in tonnes) and unit water footprint data (in cubic meters per tonne). The first remarkable database of uWF data has been prepared and shared by the Water Footprint Network, which published a large open-access dataset of uWF for several primary and processed agricultural goods, having crop and animal origin (Mekonnen & Hoekstra, 2010a, 2010b). This database, named WaterStat, includes average values over the period 1996-2005 and has been the basis of the water footprint assessment as presented, e.g., in Hoekstra et al. (2011). Other uWF datasets exist, which are based on spatially distributed models coupling the soil water balance with vegetation growth (see, e.g., Tuninetti et al., 2015, and references therein); such databases mostly refer to a single year or a period or to long-term averages. Other datasets, referring to blue water or to scarcity-weighted indicators, are also available from the literature related to the Life Cycle Assessment (e.g., Pfister et al., 2011; Pfister et al., 2016). The temporal variability of uWF has been seldom considered. Few examples include water scarcity indexes (e.g., Pfister & Bayer, 2014), or annual time series of uWF in the EORA database, based on assumptions about the economic growth of different production sectors (Lenzen et al., 2013). Recently, Tuninetti et al. (2017) proposed a Fast-Track method to estimate annual uWF values from WaterStat using agricultural yield data.

International trade statistics of agricultural goods are organized and shared by, e.g., the Food and Agriculture Organization (FAOSTAT) and the United Nations (UN-COMTRADE). Early publications by the Water Footprint Network (e.g., Hoekstra & Chapagain, 2011) are based on the combination of such trade databases and WaterStat to produce WF assessments. Trade data are also organized and shared as Input-Output tables, tracing supply chains across sectors and countries, whose worldwide dimension is captured by global multi-regional input-output tables (MRIO) (see Tukker & Dietzenbacher, 2013, for a review). In such a framework, some MRIO databases offer specific water-related extensions, quantifying water volumes associated to international trade (e.g., Geschke & Hadjikakou, 2017). Two

relevant examples are the EORA database (Lenzen et al., 2013) and EXIOBASE (Stadler et al, 2108), both including a water assessment distinguishing between green and blue water and including the temporal variability, although product categories and geographical regions are more aggregated than in the present study. Supply chains and trade of specific products, with their impact on the local environment and the water resources are also the objectives of the TRASE project developed by the Stockholm Environment Institute and the Global Canopy Programme (SEI, 2019). Such

project focuses on a limited set of products, although accurately investigating their supply chain and environmental effects.

Methodologies for VWT and WF assessment can be classified in two approaches: the bottom-up and the top-down approach. The *bottom-up* approach refers to a process-based analysis, with a detailed description of production

processes and associated water volumes. Within such approach, the uWF of each good is multiplied by the (produced or traded) quantity of such good and resulting water volumes are then summed across goods. WaterStat is the main example of a bottom-up approach The *top-down* approach aims at tracing full supply chains throughout economic sectors and different countries. Input–output analyses, frequently used in Economics for environmental assessments, belong to this approach (Duarte & Yang, 2011). Bottom-up approaches do not consider the entire supply chain of goods

and can be affected by truncation errors when used to assess the water footprint of final consumption (Feng et al., 2011). At the same time, bottom-up techniques can offer high commodity resolution considering the water associated to the production of a large variety of single (agricultural) products. A major problem affecting bottom-up approaches is the identification of the geographic origin of produced goods (Hubacek & Feng, 2016). In many cases, product re-export disconnects producing and consuming countries, now allowing a correct identification of dependencies and

externalities. In the present work, we improve the traditional bottom-up approach by identifying the origin of produced goods and reconstructing the supply chain of agricultural goods, implementing the method proposed in Kastner et al (2011). With such improvement, the VWT quantified in this study aims both at best estimating the water embodied in bilateral trade and at providing accurate estimates of the total virtual water embedded in final consumption (Feng et al., 2011; Lenzen et al., 2013).

In this publication, we present an open-access database of virtual water trade, including the annual trade matrices (years 1986-2016) and the annual virtual water export (years 1961-2016) associated to a large number of agricultural products, as well as their unit water footprint in all countries (years 1961-2016), referring to the sum of green water (originated from rainfall) and blue water (originated from surface- and ground-water bodies). Starting from the uWF dataset in

Mekonnen & Hoekstra (2010a, 2010b), we extend it to provide annual statistics of uWF. Improvements also include the

differentiation between the production-side and supply-side of uWF. The new time-varying uWF are applied to the FAOSTAT datasets of agricultural production and trade. The results of this analysis constitute the CWASI database.

The database addresses several needs: (i) the need for a comprehensive database of uWF, WF and VWT, (ii) the need to
adopt unit water footprints that vary in time, as recently pointed out by D'Odorico et al. (2019), (iii) the need to disentangle the production-side and the supply-side uWF to coherently assess the WF of production and consumption, (iv) the need for ready-to-use detailed trade matrices, accurately tracing goods' trade and origin, suitable for network analyses. The uWF dataset may also be useful for other methodologies of WF and VWT assessments, such as those based on input-output matrices or the one proposed in the ISO standardization (ISO, 2014).

The present database has been developed within the EU-funded CWASI project "Coping with WAter Scarcity In a globalized world" and it is shared through an online open-access repository (Tamea et al., 2020). In a relatively recent overview of the field, the research lines that originated from the concept of WF were identified (Hoekstra, 2017). These are the role of trade and globalization in goods production and consumption and how they affect local water issues, the
comparison of water requirements with water availability and renewability, and the supply-chain approach applied to water management. With the CWASI database we aim at contributing to these research lines and provide all researchers with an up-to-date and ready-to-use starting point for their research. The database will welcome additions and external contributions that may possibly become available in the future and will represent an open and shared source of data on water footprint and virtual water trade.

**2 Data and preliminary arrangements**

From FAOSTAT, the statistical database of the Food and Agriculture Organization (FAO), we collected 31 years (1986-2016) of trade data of agricultural goods (FAO, 2019b). Data originate from national accountings and are available as records containing the following information: reporting country (with FAO code), type of trade (import or export), partner country reported within the trade record (with FAO code), year, commodity (with FAO code), unit of
measure, quantity. From FAOSTAT, we also collected 56 years of agricultural production data including crop-based and animal-based commodities, containing this information: producing country (with FAO code), year, commodity (with FAO code), unit of measure, quantity (FAO, 2019a, 2020a, 2020b, 2020c, 2020d). From the same source, data of agricultural yield and harvested area were also collected for each considered crop, country and year in the period 1961-2016 (FAO, 2019a). Reference unit water footprint values for every commodity and country, averaged around the year
2000 (1996-2005 period), are taken from WaterStat (Mekonnen & Hoekstra, 2010a, 2010b), as well as the product

fraction and the value fraction needed for the computation of the uWF of processed crops. A detailed summary of data sources has been arranged in Table 1.

## 2.1 Commodities

Production and trade data collected from FAOSTAT include crops, processed crops, livestock primary, livestock processed and live animals. The commodities currently included in the CWASI database are 357 and have been identified as those whose FAO code or name or description could be associated to a WaterStat database entry (commodities are listed in the Appendix, Table A.1). Commodities includes all products in the "Crop" production statistics of FAO, many processed crops with the exception of feed products (such as bran and cake), animals and

animal-based products for most relevant species. Among all commodities, some appear in both trade and production data, some appear only in trade and some other appear only in production. Production data are only available for primary goods and for few processed goods, while trade includes primary and a larger set of processed goods. For example, the flour of wheat or the bread are only available as trade data because production data only include the primary commodity (wheat). Conversely, yams or sugar cane are only available as production data because their trade is

not recorded in the FAO statistics, possibly because they are not internationally exchanged as raw product. Commodities have been subdivided into 9 categories whose numbers of produced and traded commodities are specified in Figure 1. The FAOSTAT database provides for each commodity and year the amounts of goods produced (or traded) in any given country (or pair of countries) expressed in tons or heads, depending on the type of product (see the details in Table 1).

## 2.2 Countries

The database considers all geographical/political/economical entities reporting (or reported for) at least one product and one year, either in the trade or the production data. From 1961 to 2016, agricultural goods were produced and traded among 255 entities having a temporary or permanent activity (the full list is reported in the Appendix, in Table A.2). Not all the 255 countries were active along the whole considered period, as they underwent political-administrative

changes. Examples include is the collapse of the USSR, the separation of Eritrea from Ethiopia, or the splitting of Belgium and Luxembourg, which were considered a single entity until year 2000. Despite being inactive, a country may be reported by partners as importing or exporting goods. Values reported for a country outside its range of active years are associated to the corresponding active country or to the largest of them (e.g., a trade reported towards USSR in 1992 is associated to the Russian Federation). The following non-overlapping FAO entries, "China, Mainland", "China, Hong

Kong SAR", "China, Macao SAR", "China, Taiwan Province of", have been considered in place of the aggregate entry

"China". Two entries of unclear location (Neutral Zone, Unspecified) are listed but values are not considered, in order to avoid the erroneous accounting of trade fluxes. Discontinuities in the active periods each country are listed in the Appendix, in Table A.2.

### 2.3 Trade matrices

The detailed trade data provided by FAO (2019b) include the international trade records reported by each country. Reporting countries across the years are 186, whereas the remaining ones (up to 255) are only reported by others. There is a total of 9 million records (i.e., trade flows per country pairs, per commodity and per year, for the commodities included in the CWASI dataset) and the number of records reported by each country is detailed in Figure 2. These records are used to reconstruct the trade matrix $M$ for each commodity and year, having dimensions 255 x 255 and

carrying the exporting countries on the rows and the importing countries on the columns. The matrix element $M(i,j)$ thus identifies the trade flow from country $i$ to country $j$, which is clearly different than the flow from country $j$ to country $i$, i.e. $M(i,j) \neq M(j,i)$. Sub-national trade is not considered in these matrices and the terms on the diagonals are zeros.

    A problem arising in the construction of trade matrices is that the same trade flow can be reported twice in the

FAOSTAT database, once by the exporting country and once by the importing country. When a trade flow is reported by only one of the two countries, the reported flow is used to construct the matrix (single record); this is the case for 40% records in the database. All other records are "double" (reported twice) and require a comparison between the declarations of the exporting and the importing countries, which are usually different, with a mean (absolute) relative difference, across all goods, countries and years, of 61%.

The choice of a value from two double records is called "reconciliation" and the method here adopted is based on the identification of the most reliable reporting country among the two involved in each flow, and the use of the flow being reported by it. The reliability of countries is measured per commodity and per year with a data-based approach detailed below and adapted from Gehlhar (1996).

### *Country reliability*

For each product, $p$, and year, $t$, two trade matrices are built, one matrix collecting all "Importer-Reported" flows and the other matrix collecting the "Exporter-Reported" flows. The matrices have the same structure and dimensions, with the exporter countries on the rows and the importing countries on the columns. Then a reliability index is calculated for each country, $c$, differentiating between import and export.

First, an accuracy measure ($A$) is defined for every flux, from country $i$ to country $j$, as

$$180 \quad A(i,j) = \frac{|IR(i,j) - ER(i,j)|}{max\{IR(i,j), ER(i,j)\}}, \tag{1}$$

with $IR(i,j)$ being the importer-reported trade flux and $ER(i,j)$ being the exporter-reported flux. The measure is modified from Gehlhar (1996) to maintain the conceptual symmetry between import and export. The smaller is the measure and the more similar is the information reported by the importing and exporting country.

Then, the reliability of each country is measured, separately for import and export, based on the comparison between the flows reported by the country and by its trade partners. For every country, $c$, the reliability index for imports, $RI_{imp}(c)$, and for exports, $RI_{exp}(c)$, are defined as follows:

$$RI_{imp}(c) = \frac{\sum_j^{acc} IR(j,c)}{\sum_j^{all} IR(j,c) - IR(w,c)}, \quad RI_{exp}(c) = \frac{\sum_i^{acc} ER(c,i)}{\sum_i^{all} ER(c,i) - ER(c,w)}. \tag{2}$$

where $IR(j,c)$ is the flux from country $j$ to $c$, as reported by $c$ (importer-reported), and $ER(c,i)$ is the flux from $c$ to any country $i$, as reported by $c$ (exporter-reported) respectively. $\Sigma^{all}$ is the sum of all import or export fluxes reported by $c$ and $\Sigma^{acc}$ is the sum of acceptable fluxes only, defined as the fluxes whose accuracy $A$ (Eq. (1)) is smaller than an acceptance threshold, set to 20% as in Gehlhar (1996). $IR(w,c)$ and $ER(c,w)$ in Equation (2) are, respectively, the import from, and the export to, the worse partner $w$ defined as the ones having the maximum (worse) flow-weighted accuracy measure ($WA$) defined, for import and export fluxes, as

$$WA_{imp}^c(j) = A(j,c) \cdot \frac{IR(j,c)}{\sum_j^{all} IR(j,c)}, \quad WA_{exp}^c(i) = A(c,i) \cdot \frac{ER(c,i)}{\sum_i^{all} ER(c,i)}. \tag{3}$$

$A(j,c)$ is the accuracy level of flux $IR(j,c)$ and $A(c,i)$ is the accuracy level of flux $ER(c,i)$; the denominators are, respectively, the sum of all imports and all exports reported by country $c$.

Reliability indexes are calculated by country, commodity, year and flow direction (import and export). This because the reliability of a country in reporting import and export may be different, the attitude of a country to over-report or under-report may differ by products, e.g. depending on taxation, and the reliability of a country may change in time, e.g. according to socio-political factors. The direction- and commodity- averaged $RI$ of reporting countries are shown in Figure 2 with the darker (lighter) line corresponding to the newest (oldest) values. Countries more involved in trade and reporting more information (to the left) are characterized, on average, by a larger reliability, while countries less involved in trade have lower average reliability, which used to be very low in the past. Current $RI$ values, instead, are more uniform across countries.

Having computed all reliability indexes, the "reconciled" trade matrix for each good and year is built, combining importer-reported and exporter-reported data. Each matrix element $M(i,j)$ is taken from the $IR$ or $ER$ matrix if the importing country $j$ or the exporting $i$ has a larger reliability index respectively. Where the reliability indexes are equal, the country having larger acceptable fluxes is chosen.

## 3 Unit water footprint

The unit water footprint measures the amount of water required to produce a unit amount of product and it can be expressed as $m^3$/ton or, equivalently, as l/kg. The present work considers the sum of green water (originated from rainfall) and blue water (originated from surface- and ground-water bodies). Depending on the type of commodity, different approaches are applied for the computation of the unit water footprint. In the present work we propose a differentiation between the uWF of production (uWFp) and the uWF of supply (uWFs). The uWFp refers to locally-produced crops whose water footprint depends on the crop actual evapotranspiration and crop yield, with annual estimates starting from 1961. The uWFp is a suitable indicator to assess the WF of agricultural production. The uWFs, instead, refers to the domestic supply, which relies both on local production and on international trade. Country-scale domestic supply is available for human consumption, food manufacturing, feed for livestock and as export towards other countries. The impossibility to track local production and imports into consumption and exports, within each country, makes the uWFs the best indicator to be used in conjunction with consumption and export data. The uWFs is computed averaging local production and imports, after having identified the countries of origin of the goods with an appropriate procedure applicable from year 1986.

For primary crops, it has been possible to estimate both the uWFp and the uWFs. Processed crops are produced from a root product which may or may not originate from local production. The absence of systematic FAO data about the production of processed crops prevents the differentiation between the unit water footprint of production and of supply. Therefore, processed crops considered in this study will have a single unit water footprint, depending on country and year, computed from the uWFs of the root product. Finally, animal-based products are here considered only with the WaterStat values, without temporal variability.

### 3.1 Unit water footprint of locally-produced primary crops in time

When considering the production of primary crops, the unit water footprint of production, uWFp, is a function of the actual evapotranspiration along the growing period of the crop and the crop actual yield. Due to precipitation, evapotranspiration, and yield fluctuations, the uWFp exhibits significant spatio-temporal variability. We computed the uWFp in a given year by means of the Fast-Track (FT) method, introduced and substantiated in Tuninetti et al. (2017). This method is based on the use of the WaterStat database (Mekonnen & Hoekstra, 2010a, 2010b) for expressing the spatial variations of evapotranspiration and on a ratio of agricultural yields for expressing the temporal variability of the unit water footprint, not detailed in WaterStat.

According to the Fast-Track method, the unit water footprint of an agricultural product $p$ produced in country $c$ in year $t$, i.e. $uWFp_{c,p,t}$, reads:

$$uWFp_{c,p,t} = \overline{uWFp_{c,p,T}} \cdot \frac{\overline{Y_{c,p,T}}}{Y_{c,p,t}}, \tag{4}$$

where $\overline{uWFp_{c,p,T}}$ is the reference unit water footprint provided by WaterStat (Mekonnen & Hoekstra, 2010a, 2010b)
corresponding to an average in the period $T$=1996-2005, $\overline{Y_{c,p,T}}$ is the average crop yield over the same period $T$, and $Y_{c,p,t}$ is the annual crop yield in a generic year $t$ in the range 1961-2016. The average crop yield is obtained as an average of the annual yields in the years 1996-2005, weighted by the harvested areas across the years in country $c$, based on FAOSTAT data (FAO, 2019a).

The Fast-Track method keeps implicitly constant the actual evapotranspiration of crops, equal to the long-term average used in the WaterStat statistics, but this hypothesis should come at no surprise. On the one hand, yield implicitly expresses many factors, including climatic conditions, water availability, soil fertility and agricultural practices among others, and yield temporal variations dominate over the variability of the water volumes used (evapotranspired) by crops. On the other hand, the uWF is less sensitive to hydro-climatic conditions than actual evapotranspiration, because
it is defined as the ratio between evapotranspiration and yield, both reacting with equal signs to hydro-climatic fluctuations (see, e.g., Doorenbos et al, 1979). Additional indications about the uncertainty associated to the Fast-Track method are provided in Section 5.1.

### 3.2 Primary-equivalent trade matrix

For the correct identification of countries of origin of the crops traded internationally, the reconstruction of a primary-
equivalent trade matrix, $\boldsymbol{M}_{eq}$, is necessary (Kastner et al., 2011). This is defined as

$$\boldsymbol{M}_{eq} = \boldsymbol{M}_p + \sum_{dp}\left(\boldsymbol{M}_{dp} \cdot \frac{f_v}{f_p}\right), \tag{5}$$

where $\boldsymbol{M}_p$ is the trade matrix of any root-product, $\boldsymbol{M}_{dp}$ is the trade matrix of the derived products ($dp$) and $f_p, f_v$ are the product fraction and value fraction which convert the derived products into a root-product equivalent quantity. The summation is extended to all derived products which originate from the same root product and, in the case of a multi-
step supply chain, Eq. (5) is applied iteratively until reaching a root product that is also a primary crop. The product fraction, $f_p$, is defined as the weight of a derived product obtained from a ton of input product. For example, a ton of nuts with shells leads to $f_p$ (<1) tons of shelled nuts. The value fraction, $f_v$, is the market value of the derived product

divided by the aggregated market value of all derived products resulting from a ton of input product. For example, in a production process of wheat flour there are other economically valuable by-products (e.g., wheat germs to feed animals); hence, the value of wheat flour constitutes only a portion (i.e., the value fraction) of the total value generated by the process. Product fractions and value fractions used in the CWASI database are time- and space- invariant and are taken from Mekonnen & Hoekstra (2010a, 2010b), as well as the root products and the full supply chains of the considered commodities.

### 3.3 Supply-side unit water footprint of primary crops

The country supply of a primary crop results from the sum of local production and imports, where imports may occur from producing or non-producing countries, the latter case testifying a re-export of goods produced elsewhere. Therefore, the unit water footprint of supply, uWFs, is proportionally contributed by local production and by trade, specifying the relative contribution of every country from which the goods originated from, considering re-exports and the processing of goods, if necessary. For each primary-equivalent crop and each year, we can define a column vector, $S$, containing the supply of all countries as rows. This vector is calculated as the sum of the production vector, $P$, and of the imports obtained from the bilateral trade matrix $\boldsymbol{M}_{eq}$, where $M_{eq}(i,j)$ identifies the trade flow from $i$ to $j$ as

$$S = P + \boldsymbol{M}'_{eq} \cdot I \,, \tag{6}$$

where $I$ is a column vector of ones (i.e., a summation vector) and $\boldsymbol{M}'_{eq}$ is the trade matrix transposed. Hence, the uWFs of a country depends both on the domestic uWFp (through $P$) and on the uWFp of the origin countries, where the product is produced.

In order to trace the actual origin of the country's supply, namely tracing its origin back to the country where it was produced, we adopt the approach proposed by Kastner et al. (2011). First, we define a matrix $\boldsymbol{R}$, where each element $R(i,j)$ is the quantity of supply in country $i$ that is produced in country $j$. A first approximation of $\boldsymbol{R}$ can be based on reported flows only, and being equal to the sum of a diagonal matrix with elements of the $P$ vector on the diagonal, i.e. $diag(P)$, and the transposed trade matrix, $\boldsymbol{M}'_{eq}$. However, this approximation misses the fact that exporting countries may obtain the exported products not only from local production, but also from import. To account for this fact, a matrix of export shares, $\boldsymbol{X}$, can be defined as

$$\boldsymbol{X} = \boldsymbol{M}'_{eq} \cdot diag(S^{-1}) \,, \tag{7}$$

where $X(i,j)$ is the share of country $j$'s supply that is exported to country $i$. The term $diag(S^{-1})$ denotes a diagonal matrix made up by the reciprocal elements of $S$. In turn, the imported and re-exported products may partly originate

from local production and import, and so on, recursively. It has been shown by Miller and Blair (2009), that such procedure converges to

$$R = (I - X)^{-1} \cdot diag(P),$$ (8)

where the $R$ matrix identifies where the supply of each country originates from and $I$ is the identity matrix. For further details and exemplification, see Kastner et al. (2011).

By knowing the uWFp of the primary crop in such countries, we can now define the unit water footprint of supply in country $c$ and year $t$ of the primary product $p$, i.e. $uWFs_{c,p,t}$, as

$$uWFs_{c,p,t} = \sum_{j=1}^{255} uWFp_{j,p,t} \cdot \frac{R_{p,t}(j,c)}{S_{c,p,t}}.$$ (9)

The evaluation of uWFs corresponds to a weighted average of the uWFp values, where the weights are the actual fractions of supply, $S$, traced back to their origins. Eq. (9) is valid for every primary crop $p$ and year $t$, considering that trade matrices, production vectors and uWFp values change from year to year. It is worth noticing that because the trade matrices are available form 1986 only, the uWFs can be built from that year only.

### 3.4 Unit water footprint of processed crops in time

Processed crops are based on the processing of root products, which are available as country's supply. The time-varying unit water footprint thus depends on that of the root product and on the conversion factors, i.e. (Hoekstra, et al., 2011),

$$uWF_{c,dp,t} = uWFs_{c,p,t} \cdot \frac{f_v}{f_p},$$ (10)

where $uWF_{c,dp,t}$ is the unit water footprint of the processed crop (or derived product, $dp$), $uWFs_{c,p,t}$ is the unit water footprint of supply of the root product from which it derives ($p$), $c$ and $t$ are country and year, respectively, $f_p$ is the product fraction and $f_v$ the value fraction of the processed crop (see Section 3.2). The method takes into account the temporal variability associated to both the crop production, through the Fast-Track method applied to the primary crop, and the evolution of trade, through the Kastner's method applied to the crop supply; the method does not include water inputs for processing of goods. When supply chains are formed by multiple steps, for example in the case of bread, made with flour, made in turn with wheat, Eq. (10) is applied routinely at each step. Within the CWASI dataset, the longest supply chain is made of 4 steps, leading to final products such as refined sugar or chocolate.

Equation (10) describes the unit water footprint without differentiating between production and supply. This is because the absence of FAOSTAT data of production of most processed crops (FAO 2020a) hinders the application of the Kastner's method (Sect. 3.3), thus an explicit accounting of countries of origin of trade, as in the case of primary crops.

However, the trade of processed crops is implicitly taken into account in the procedure, thanks to the use of the primary-equivalent trade matrix (Eq. (5)) which serves to compute the uWFs of the primary crop.

For the very few derived products without indication of the root product (Mekonnen & Hoekstra, 2010a), an association is made which is based on logical considerations (such as "Figs dried" deriving from "Figs") or on similitudes of products. The "Sugar" products (Raw Sugar, Refined Sugar,...) were also missing the root-product, likely due to a lack of information. For these products, we have traced back the root product to the product most largely available as country supply (either "Sugar Beet" or "Sugar Cane").

### 3.5 Unit water footprint of animal-based commodities

Animal-based commodities considered in FAOSTAT are grouped in three categories: "Live animals", "Livestock primary", and "Livestock processed" (FAO, 2020b, 2020c, 2020d). Products of the first category are given in Heads unit, which have been converted in tons according to FAO conversion factors (FAO, 2013). The missing conversion values for some countries (or animal products) have been assigned with an average value by category or considering similar animals. Due to the lack of reliable data about country-specific animal diets and their temporal variability as well as the lack of detailed trade matrices of feed crops, we do not currently provide a time-dependent unit water footprint for the animal-based commodities. Nevertheless, we include these products in the present database adopting the country-specific values provided by WaterStat (Mekonnen & Hoekstra, 2010b) and without differentiating between production and supply (i.e., $uWFp = uWFs$). These values of unit water footprint take into account the feed-animal-commodity global supply chain, considering locally-produced and imported feed (Mekonnen & Hoekstra, 2012) but are only available as time-averaged values over the period 1996-2005. Here data are generically referred to year 2000 and are arranged consistently with the rest of the CWASI database.

### 4 Virtual water trade and water footprint indicators

### 4.1 Water footprint and VWT data

The water footprint of agricultural production in a country and year is obtained by multiplying the production data (FAO, 2019a, 2020a, 2020b, 2020c, 2020d), expressed in metric tons, by the corresponding (commodity, country, year) unit water footprint, considering the unit water footprint of production, $uWFp_{c,p,t}$, in the case of primary crops. A problem arises when a country was not a producer in the 1996-2005 decade, thus it does not have an associated value in the WaterStat database. In such case, the uWF in the closest producing country within a certain distance (10°) is taken; if no producing countries are found (e.g., in the case of remote islands, or small producing areas), then the global

average weighted by production is used. In the case of countries having experienced political discontinuity, for example belonging to a larger country before years 1996-2005 considered in the WaterStat database (e.g., USSR), the reference value of uWF required in Eq. (4) is computed as a production-weighted average of the values of countries belonging to the union and available in WaterStat. This average value is then used to reconstruct the annual uWF from 1961 up to the year of the disaggregation. After converting the agricultural production into water volumes, the overall water footprint of production is obtained by summing across all commodities. Care must be used to avoid double-accounting of water footprints of primary and derived goods. For this reason, only primary products must be considered in aggregated production data. In particular, when dealing with animal-based commodities, one should avoid the inclusion of both livestock and the corresponding products as well as the crops used to feed the livestock. Primary, or single-accounting, products to be included in the sum are indicated in the Appendix, in Table A.1.

Computation of the supply-side unit water footprint of goods enables the fast computation of the water footprint associated to the consumption of commodities, under the hypothesis that consumption (and export) shares with the country's supply the same mix of local and imported goods. The water footprint of consumption in a country and year can thus be obtained by multiplying the consumed quantity of each good by the unit water footprint of supply, $uWFs_{c,p,t}$ (per commodity, country and year), then summing across all commodities. In this case, there is no double-accounting issues.

The virtual water trade is obtained by multiplying trade data (FAO, 2019b), expressed in metric tons, by the unit water footprint of supply, $uWFs_{c,p,t}$ of the exporting country. Thanks to the new definition of supply-side unit water footprint, this computation allows one to take into account the origin of goods, which are traced back to their origin countries along the supply chain. In the few cases of goods exported from countries not having an associated uWF (less than 1% of trade links over the whole period, mainly from minor countries or remote islands), the global average uWF of supply is used, weighted by all countries' exports. Virtual water trade associated to animal-based commodities is given for year 2000 only, consistently with their unit water footprint, with the uWFp used for the conversion, not being available a uWFs for these commodities yet.

## 4.2 The uWF index

In the Results section, the (volumetric) water footprint and the virtual water trade are summed across different commodities and the overall trends are assessed in time. However, the unit water footprint of different commodities cannot be assessed as a whole, but only for one commodity at a time. To overcome such problem, an appropriate index

is constructed in analogy to some economic indices aggregating prices of different commodities, such as the Agriculture Producer-Price Index (in FAOSTAT) calculated with the Laspeyres approach. The index is built as the inverse ratio between the WF of production (in m$^3$) of all commodities ($i$) in all countries ($c$) in year 2000 and the WF obtained with the same quantities (year 2000) but with uWF in year $t$, i.e.

$$I(t) = \frac{\sum_{i,c} \; uWF(i,c,t) \cdot P(i,c,2000)}{\sum_{i,c} \; uWF(i,c,2000) \cdot P(i,c,2000)} \cdot 100. \tag{11}$$

In such way, $I(t)$ expresses the variation of uWF across all agricultural commodities, weighted by the productions in year 2000, $P(i,c,2000)$. A similar index as in Eq. (11) can also be built for trade using, e.g., the exports of each country in year 2000 as weights, thus leading to a uWF index for trade. In addition, indexes (for production or trade) referring to single categories of goods can be built by aggregating only the goods belonging to a given category.

## 5 Results

The importance of considering a time-dependent unit water footprint is highlighted in Figure 3, which shows the temporal trends of the global average uWF of production of some commodities. The global average is computed by weighting the uWFp of each country by the country production of such crop. The relevance of the temporal change is evident, ranging from 4000 to 1500 m$^3$/ton over the whole considered period. It is worth noticing that the uWF od production of other major crops are shown in Tuninetti et al. (2017).The values considered in WaterStat refer to the period $T$=1996-2005, highlighted by a grey shade in Figure 3. It is clear that the average value in such reference period is scarcely representative of the whole period considered in the present dataset. It is thus very important to consider the temporal variability of unit water footprint, especially in analyses spanning long periods or periods different than years 1996-2005.

The temporal variation of the uWF of production of crops is marked all over the world. If compared to the values averaged over the period 1996-2005 (as in the WaterStat database), the uWFp computed with the Fast-Track method at the beginning and at the end of the considered period are very different. Figure 4 shows the relative change of the uWFp of wheat in 1961 and 2016 with respect to the 1996-2005 average. The variation is quite uniform worldwide with improvements (decreases of uWF) in both periods which are consistent with the different duration of the two periods. Extreme variations have occurred in China (largest improvement from 1961) and in African countries, showing large improvements in time but also occasional worsening due to unstable socio-economic conditions. It should be noticed that few countries worldwide do not produce wheat or miss FAOSTAT or WaterStat data: in such cases, uWFp values from nearby countries or worldwide averages are used instead. Having a uWF, for any good, in all countries and years

is a need dictated by the chance that a non-producing country is an exporter (or re-exporter) of such good and thus requiring a uWF for the conversion of the trade flow into virtual water.

A comparison between the uWF of production and supply of primary crops is very informative. Figure 5 highlights the
absolute difference for wheat and soybean with red colours indicating countries where the uWFs is smaller than uWFp and green colours for the contrary. The more intense is the red colour, the more efficient is the crop import in saving global water resources because the imported crops are produced with lower uWF that the local uWF of production. This is the case for several African countries, some South-American ones and Thailand for wheat and several South-Asian countries for soybeans. On the contrary, the more intense is the green colour, and the more efficient the global
production is, as compared to imports. The extreme case of non-producing (but importing) countries is highlighted by bold contours. This is observed in several Far-East countries for wheat and by most African countries for soybeans.

Considering all commodities together, the analysis of temporal evolution requires the use of a uWF index (Eq. (11)). The index built weighting by agricultural production decreases monotonically in time (Figure 6, left), being at +50% in
1961 and -7% in 2016. The trend is less marked than in Figure 3 because all goods, and not only wheat, are being considered in the index. The uWF index weighted with exports decreases even more starting with a +85% in 1961. The difference may be interpreted in two ways, the first being that in the past times, for a given good, the production in exporting countries was more efficient than in the other (producing but not exporting) countries. The second interpretation focuses on a given country and highlights changes in the relative composition of export, with increasing
shares of water-efficient goods at the expenses of water-inefficient ones. It is worth noticing that the uWF index is built with all commodities, including those not having a uWF varying in time (e.g., animal-based products, as made explicit in Figure 6, right): considering such contribution, the index temporal variation would be even more marked. The uWF indexes by category, shown in Figure 6 (right), allows one to find similarities and differences. In time, traded fruits and vegetables have improved their uWF more than global production, with large discrepancies in the past between trade-
weighted and production-weighted indexes. Traded cereals, especially in the past, where produced with high efficiency, while in more recent years the average uWF of traded cereals became larger than that of the global cereal production. This fact may be due to the increased participation in the international trade of developing countries (Carr el at., 2013), countries producing cereals with lower efficiency than major exporters (eg., the USA). Seeds/oils and luxury food had intermediate periods where trade was on average more efficient than global production, while non-edible goods are
currently traded with low efficiency, i.e. the trade-weighted uWF index is larger than the production-weighted one.

The time-varying uWF in the CWASI database is used to assess the temporal evolution of virtual water trade across the years, considering the contribution of different categories of goods. Figure 7 updates previous versions published in the literature (e.g., Konar el at., 2011; Carr el at., 2013; Tuninetti et al., 2017) by either introducing the temporal variability

of the uWF of crop-based goods, expanding the number of considered crops and/or extending the temporal range considered. Total VWT has increased from 750 to 2400 $km^3/y$ in the considered period (about 1000 $km^3/y$ in 1986). Major categories are cereals, luxury food, seeds/oils and vegetables, with the relative contribution of cereals, which was very large in the 60ies, being outperformed by the other categories in the most recent years. VW volumes associated to cereals has doubled in the considered period, while volumes associated to vegetables has grown 9-fold. The growth of

animal-based products is remarkable, but it should be specified that it only reflects the increased trade quantity without considering the temporal variability of uWF.

### 5.1 Uncertainties and limitations

Despite the large amount of information and the many improvements provided with the CWASI database, the data uncertainty and a few cautions are worth to be mentioned. The time-varying unit water footprints of crops and crop-

455 based commodities are estimated with a simplified method (the Fast-Track method), that has been thoroughly assessed before applying it widely. For example, the Fast-Track estimates of unit water footprint were compared to the results of a complete model based on a daily soil water balance fed by year-specific hydro-climatic variables and the errors were found to be within a 10% range (Tuninetti et al., 2017). The uncertainty introduced in the unit water footprint estimates with the Fast-Track method is also lower or comparable to the model uncertainty associated to the water footprint

assessment, verified by a comparison with the WaterStat values (see Tuninetti et al., 2017).

The Fast-Track method, initially applied to 4 crops (wheat, maize, rice and soybeans), has been extended in the CWASI database to a large set of primary products, including cereals, fruits, vegetables, seeds, luxury food and non-edibles. The extension is justified by the fact that similar error ranges are expected in all crops, because water stress affects the

465 evapotranspiration of different crops in a similar way, the only difference being the phases of the growing periods affected by water stress and the crop coefficients describing the plant water requirements. Water stress is assumed not to affect irrigated crops, implying that actual evapotranspiration matches the crop maximum evapotranspiration in irrigated conditions. Uncertainty associated to the Fast-Track method has been sparsely checked on other crops than the first 4, and the range of errors found in Tuninetti et al. (2017) has been confirmed. Considering the hypothesis of a long-

470 term average actual evapotranspiration of crops, we suggest to use with care single-year data of uWF, as well as WF and VWT. It is precautionary to consider single-year data in a temporal perspective, such as a trend analysis, or use a multi-year average to minimize the error and avoid misinterpretations of year-specific results.

A minor point of caution is related to the supply-side uWF, which averages country's local production and import. This
variable is the best estimate to be used in association to countries' export and consumption, unless more detailed information is available about the origin of country's export or consumption. If local production or import should prevail, as compared to the average country's supply, a more precise weighted average of unit water footprint will be enabled by such information.

For what concerns the uWF of animal-based commodities, as well as their WF and the VWT, they are here reported for year 2000 only, referring to the average over the years 1996-2005 in the WaterStat database. Where necessary, these values have been applied to production and trade occurring in different years (see Figures 6-7) although cautions in such applications should be used.  This limitation can be overcome when reliable data on the country-specific feed composition and diet of animals  will become available along the considered time period.

**6 Conclusions**

The globalization of water resources through the international trade of food and agricultural goods is a remarkable global environmental change of our times, and the scientific community is devoting great effort to study it. The quantification of the volumes of water involved in the production and trade of agricultural goods is a key tool to investigate the water-food-trade nexus issues. This study presents an open-source database specifically developed for
this purpose. The main outcome of this study is the time-varying unit water footprint for the years 1961-2016 and the virtual water trade matrices for the years 1986-2016 of hundreds of commodities from the food and agricultural sector. The water footprint of production per commodity are also available annually in the period 1961-2016. The current database includes a total of almost 30 million data, half of them being elements of the trade matrices. The introduction of a supply-side estimate of the unit water footprint brings much more detail in the water footprint accounting. This is a
new concept and a key tool in the expedite and accurate accounting of the virtual water trade and of the water footprint of consumption. The supply-side unit water footprint overcomes previous problems related to the non-consideration of re-export and it also enables a more accurate assessment of virtual water trade, with the correct identification of countries of origin of traded goods.

The open-source database presented in this work aims to help the scientific community and policy makers to quantify and investigate the complex linkages between the global food system and water resource issues. Potential applications of the CWASI dataset range from supporting national-scale policies of water management as well as agricultural

policies oriented to the optimization of water use or, ultimately, to provide indications for price formation or for trade agreements based on the efficient and sustainable use of water resources worldwide. The CWASI database is shared through the Zenodo online open-access repository (Tamea et al., 2020) and it is planned to be improved and updated in the future, capitalizing contributions from the overall scientific community.

## Data availability

Data are available on the Zenodo repository at https://doi.org/10.5281/zenodo.3987468 (Tamea et al., 2020).

## Author contribution

ST, MT and IS analysed the data, developed the codes, and organized the dataset. All authors designed the methods and FL supervised the work, ST prepared the manuscript with contributions from all co-authors.

## Competing interests

The authors declare not to have competing interests.

## Acknowledgments

The authors acknowledge funding support provided by the European Research Council (ERC) through the project "Coping with water scarcity in a globalized world" (ERC-2014 CoG, project 647473). Information about project, data and results may be found at the web site https://watertofood.org/data. Giuseppe Zaccaria is also acknowledged for early contributions on the data collection and organization. The authors declare to have no conflict of interests.

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

**Figures**

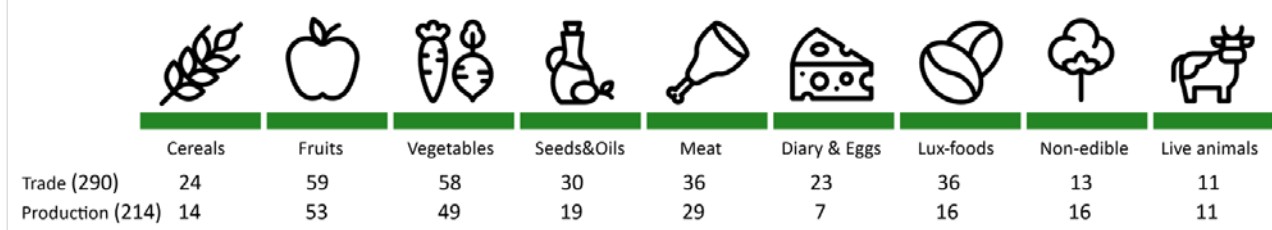

|  | Cereals | Fruits | Vegetables | Seeds&Oils | Meat | Diary & Eggs | Lux-foods | Non-edible | Live animals |
|---|---|---|---|---|---|---|---|---|---|
| Trade (290) | 24 | 59 | 58 | 30 | 36 | 23 | 36 | 13 | 11 |
| Production (214) | 14 | 53 | 49 | 19 | 29 | 7 | 16 | 16 | 11 |

**Figure 1: Commodities considered in the analysis, split into 9 categories: number of commodities in the trade and production dataset. Icons from Flaticon.com.**

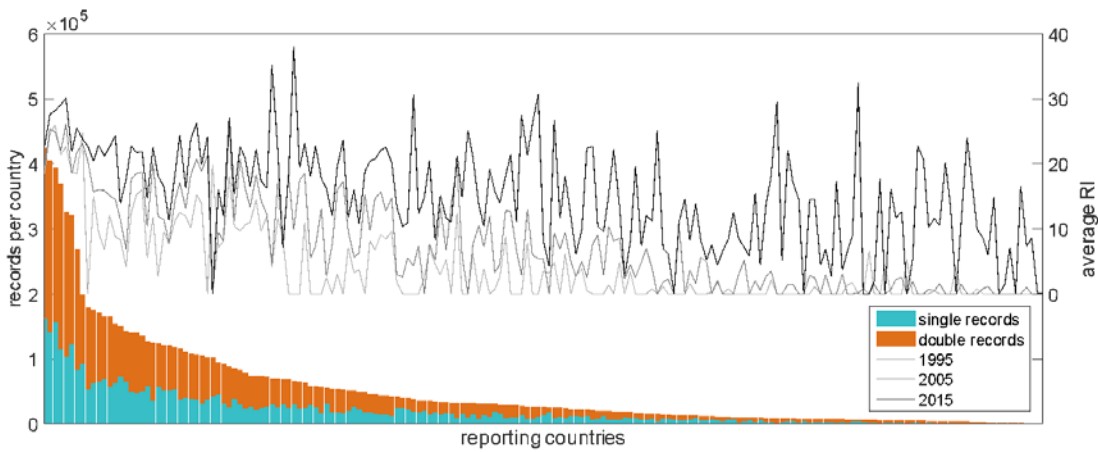

**Figure 2: Number of single and double records per reporting country (including all partners, all goods and all years). Right axis indicates the country-specific Reliability Index averaged over all goods in 3 separate years.**

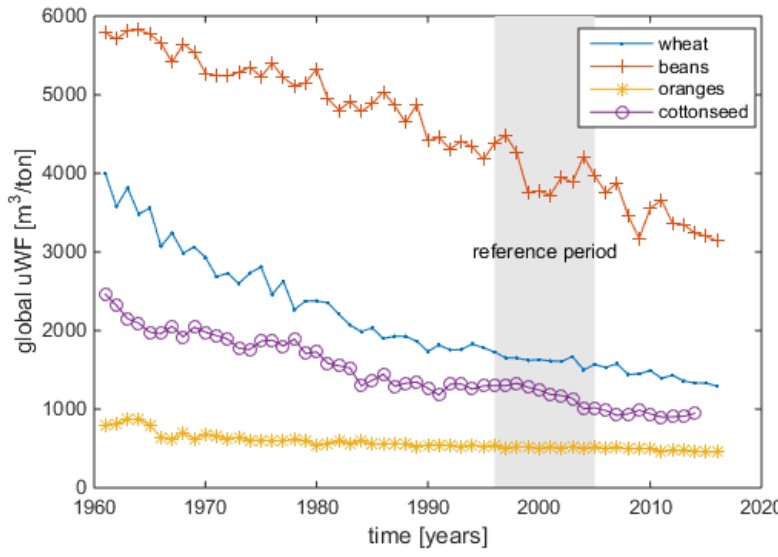

**Figure 3: Production-weighted global uWFs along the period 1961-2016 for wheat, beans, oranges and cotton.**

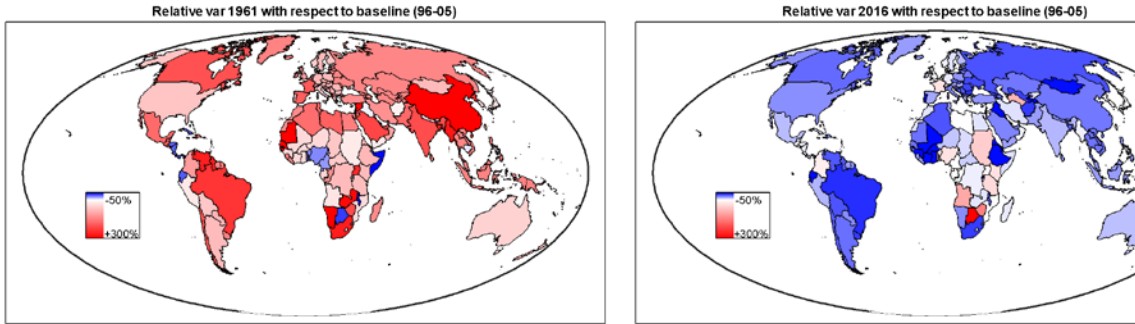

**Figure 4: Relative change (in m3/ton) in the uWFp of wheat in 1961 (left) and 2016 (right) with respect to the average in 1996-2005, using identical color ranges: red/blue colors identify higher/lower values and color intensity scales with change values. (Maps created with Matlab® R14 software, Mapping Toolbox v.2.0.3).**

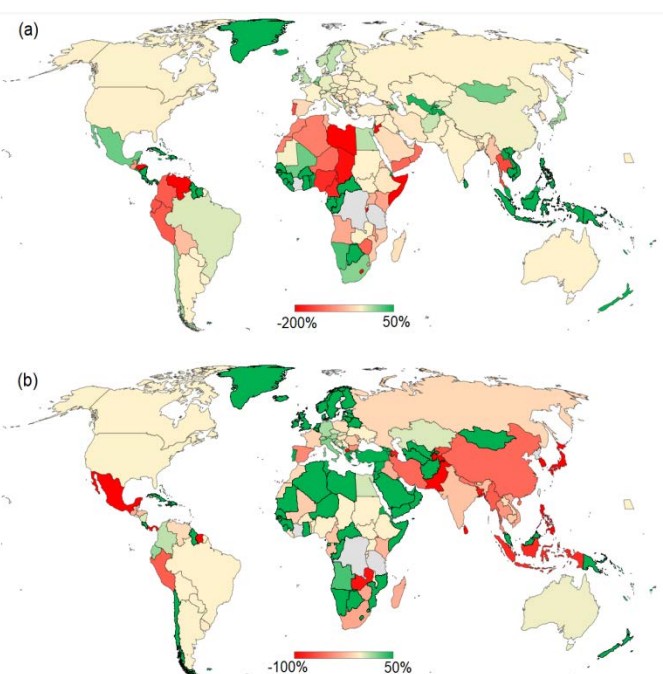

**Figure 5: Percentage difference between the uWF of production and supply of wheat (a) and soybean (b) in year 2016, calculated as the difference between uWFs and uWFp, normalized by uWFs. Bold green countries do not produce the crop; hence they only have a supply-side uWF. (Maps created with Microsoft Power Map for Excel, © Microsoft).**

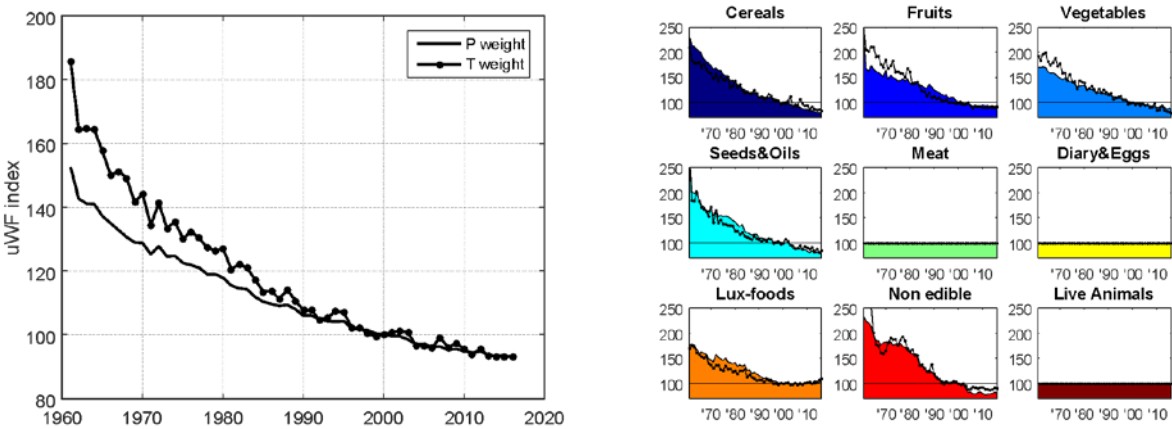

**Figure 6: Temporal variability of uWF indexes weighted with agricultural production (solid) and export (with dots) in year 2000, aggregated across all goods (left), and (right) split into the 9 categories of goods.**

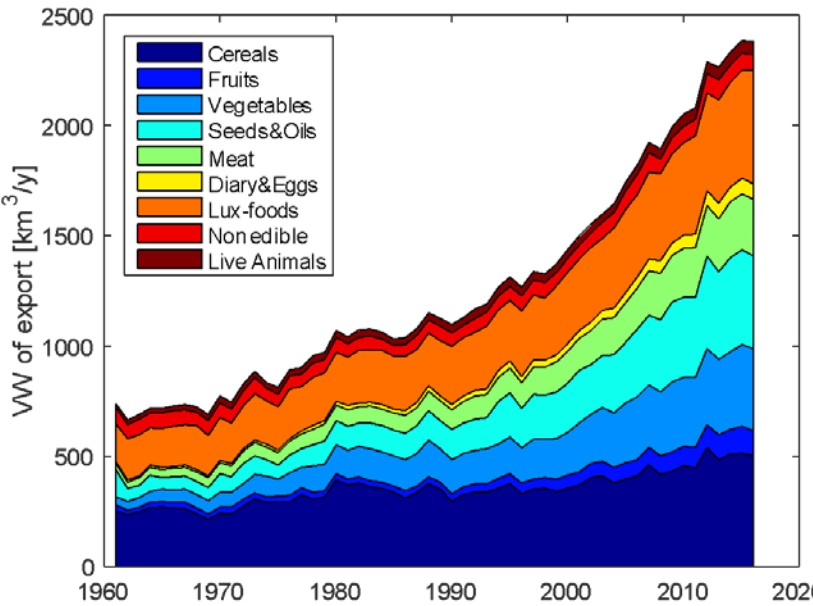

**Figure 7: Global virtual water trade (as derived from export data) from 1961 to 2016 considering the 9 categories of goods from Figure 2.**

**Tables**

**Table 1: data sources used to prepare the CWASI database.**

| Variable | Years | URL | Reference | Access date |
|---|---|---|---|---|
| Crop production, yield and harvested areas | 1961-2016 | *http://www.fao.org/faostat/en/#data/QC* | FAOSTAT (2019a) | 10/2019 |
| Production of processed crops | 1961-2016 | *http://www.fao.org/faostat/en/#data/QD* | FAOSTAT (2020a) | 01/2020 |
| Detailed trade matrices | 1986-2016 | *http://www.fao.org/faostat/en/#data/TM* | FAOSTAT (2019b) | 10/2019 |
| Animal-based primary production | 2000 | *http://www.fao.org/faostat/en/#data/QL* | FAOSTAT (2020b) | 03/2020 |
| Animal-based processed production | 2000 | *http://www.fao.org/faostat/en/#data/QP* | FAOSTAT (2020c) | 03/2020 |
| Live animals | 2000 | *http://www.fao.org/faostat/en/#data/QA* | FAOSTAT (2020d) | 03/2020 |
| Reference uWF of crop-based products | 2000 (average) | *https://waterfootprint.org/en/resources/waterstat/product-water-footprint-statistics/* | Mekonnen & Hoekstra (2010a) | 03/2020 |
| Reference uWF of animal-based products | 2000 (average) | *https://waterfootprint.org/en/resources/waterstat/product-water-footprint-statistics/* | Mekonnen & Hoekstra (2010b) | 03/2020 |

**Appendix A: Commodities and countries in the CWASI database**

Commodities included in the CWASI database are listed in Table A.1, which includes the commodity name, the FAO code, the presence of production data (1: yes, 0: no), the presence of trade data (1: yes, 0: no) and the associated category. Countries considered in the CWASI database are listed in Table A.2, which include the country name, the
645 FAO code, the position in the CWASI vectors/matrices, the indication of reporting (1) or non-reporting (0) countries, and the presence of discontinuities in the considered period.

**Table A.1** **List of commodities in the CWASI database**

| Commodity name | FAO code | in Production | in Trade | Category |
|---|---|---|---|---|
| Wheat | 15 | 1 | 1 | Cereals |
| Flour of Wheat | 16 | 0 | 1 | Cereals |
| Macaroni | 18 | 0 | 1 | Cereals |
| Bread | 20 | 0 | 1 | Cereals |
| Bulgur | 21 | 0 | 1 | Cereals |
| Rice, paddy | 27 | 1 | 1 | Cereals |
| Rice - total (Rice milled equivalent) | 30 | 0 | 1 | Cereals |
| Beverages, fermented rice | 39 | 0 | 1 | Lux-foods |
| Barley | 44 | 1 | 1 | Cereals |
| Barley Pearled | 46 | 0 | 1 | Cereals |
| Malt | 49 | 0 | 1 | Cereals |
| Beer of Barley | 51 | 0 | 1 | Lux-foods |
| Maize | 56 | 1 | 1 | Cereals |
| Germ, maize | 57 | 0 | 1 | Cereals |

| | | | | |
|---|---|---|---|---|
| Flour of Maize | 58 | 0 | 1 | Cereals |
| Maize oil | 60 | 0 | 1 | Seeds & Oils |
| Rye | 71 | 1 | 1 | Cereals |
| Oats | 75 | 1 | 1 | Cereals |
| Oats Rolled | 76 | 0 | 1 | Cereals |
| Millet | 79 | 1 | 1 | Cereals |
| Sorghum | 83 | 1 | 1 | Cereals |
| Buckwheat | 89 | 1 | 1 | Cereals |
| Quinoa | 92 | 1 | 0 | Cereals |
| Fonio | 94 | 1 | 1 | Cereals |
| Triticale | 97 | 1 | 1 | Cereals |
| Canary seed | 101 | 1 | 1 | Non edible |
| Mixed grain | 103 | 1 | 1 | Cereals |
| Cereals, nes | 108 | 1 | 0 | Cereals |
| Flour, cereals | 111 | 0 | 1 | Cereals |
| Cereal preparations, nes | 113 | 0 | 1 | Cereals |
| Potatoes | 116 | 1 | 1 | Vegetables |
| Potatoes Flour | 117 | 0 | 1 | Vegetables |
| Frozen Potatoes | 118 | 0 | 1 | Vegetables |
| Sweet potatoes | 122 | 1 | 1 | Vegetables |
| Cassava | 125 | 1 | 1 | Vegetables |
| Cassava Dried | 128 | 0 | 1 | Vegetables |
| Cassava Starch | 129 | 0 | 1 | Vegetables |
| Yautia (cocoyam) | 135 | 1 | 0 | Vegetables |
| Taro (cocoyam) | 136 | 1 | 0 | Vegetables |
| Yams | 137 | 1 | 0 | Vegetables |
| Roots and Tubers, nes | 149 | 1 | 1 | Vegetables |
| Flour of Roots and Tubers | 150 | 0 | 1 | Vegetables |
| Sugar cane | 156 | 1 | 0 | Vegetables |
| Sugar beet | 157 | 1 | 1 | Vegetables |
| Maple Sugar and Syrups | 160 | 0 | 1 | Lux-foods |
| Sugar crops, nes | 161 | 1 | 1 | Lux-foods |
| Sugar Refined | 164 | 0 | 1 | Lux-foods |
| Molasses | 165 | 0 | 1 | Lux-foods |
| Other Fructose and Syrup | 166 | 0 | 1 | Lux-foods |
| Sugar, nes | 167 | 0 | 1 | Lux-foods |
| Sugar flavoured | 171 | 0 | 1 | Lux-foods |
| Glucose and Dextrose | 172 | 0 | 1 | Lux-foods |
| Beans, dry | 176 | 1 | 1 | Vegetables |
| Broad beans, horse beans, dry | 181 | 1 | 1 | Vegetables |
| Peas, dry | 187 | 1 | 1 | Vegetables |
| Chick peas | 191 | 1 | 1 | Vegetables |
| Cow peas, dry | 195 | 1 | 0 | Vegetables |
| Pigeon peas | 197 | 1 | 0 | Vegetables |
| Lentils | 201 | 1 | 1 | Vegetables |
| Bambara beans | 203 | 1 | 1 | Vegetables |
| Vetches | 205 | 1 | 1 | Vegetables |
| Lupins | 210 | 1 | 0 | Vegetables |
| Pulses, nes | 211 | 1 | 0 | Vegetables |
| Flour of Pulses | 212 | 0 | 1 | Vegetables |
| Brazil nuts, with shell | 216 | 1 | 0 | Fruits |
| Cashew nuts, with shell | 217 | 1 | 1 | Fruits |
| Chestnuts | 220 | 1 | 1 | Fruits |

| | | | | |
|---|---|---|---|---|
| Almonds, with shell | 221 | 1 | 0 | Fruits |
| Walnuts, with shell | 222 | 1 | 1 | Fruits |
| Pistachios | 223 | 1 | 1 | Fruits |
| Kolanuts | 224 | 1 | 1 | Fruits |
| Hazelnuts, with shell | 225 | 1 | 0 | Fruits |
| Arecanuts | 226 | 1 | 0 | Fruits |
| Almonds Shelled | 231 | 0 | 1 | Fruits |
| Walnuts Shelled | 232 | 0 | 1 | Fruits |
| Hazelnuts Shelled | 233 | 0 | 1 | Fruits |
| Nuts, nes | 234 | 1 | 1 | Fruits |
| Soybeans | 236 | 1 | 1 | Vegetables |
| Soybean oil | 237 | 0 | 1 | Seeds & Oils |
| Cake of Soybeans | 238 | 0 | 1 | Vegetables |
| Soya Sauce | 239 | 0 | 1 | Vegetables |
| Soya Paste | 240 | 0 | 1 | Vegetables |
| Groundnuts, with shell | 242 | 1 | 0 | Vegetables |
| Groundnuts Shelled | 243 | 0 | 1 | Vegetables |
| Groundnut oil | 244 | 0 | 1 | Seeds & Oils |
| Cake, groundnuts | 245 | 0 | 1 | Vegetables |
| Coconuts | 249 | 1 | 1 | Fruits |
| Copra | 251 | 0 | 1 | Fruits |
| Coconut (copra) oil | 252 | 0 | 1 | Seeds & Oils |
| Cake, copra | 253 | 0 | 1 | Fruits |
| Oil, palm fruit | 254 | 1 | 0 | Fruits |
| Palm kernels | 256 | 1 | 0 | Fruits |
| Palm oil | 257 | 1 | 1 | Seeds & Oils |
| Palm kernel oil | 258 | 0 | 1 | Seeds & Oils |
| Cake of Palm Kernel | 259 | 0 | 1 | Fruits |
| Olives | 260 | 1 | 1 | Fruits |
| Olives Preserved | 262 | 0 | 1 | Fruits |
| Karite Nuts (Sheanuts) | 263 | 1 | 0 | Fruits |
| Castor oil seed | 265 | 1 | 0 | Seeds & Oils |
| Oil of Castor Beans | 266 | 0 | 1 | Seeds & Oils |
| Sunflower seed | 267 | 1 | 1 | Seeds & Oils |
| Sunflower oil | 268 | 0 | 1 | Seeds & Oils |
| Sunflower Cake | 269 | 0 | 1 | Seeds & Oils |
| Rapeseed | 270 | 1 | 1 | Seeds & Oils |
| Rapeseed oil | 271 | 0 | 1 | Seeds & Oils |
| Cake of Rapeseed | 272 | 0 | 1 | Seeds & Oils |
| Tung nuts | 275 | 1 | 0 | Seeds & Oils |
| Jojoba seed | 277 | 1 | 0 | Seeds & Oils |
| Safflower seed | 280 | 1 | 0 | Seeds & Oils |
| Sesame seed | 289 | 1 | 1 | Seeds & Oils |
| Sesame oil | 290 | 0 | 1 | Seeds & Oils |
| Mustard seed | 292 | 1 | 1 | Seeds & Oils |
| Poppy seed | 296 | 1 | 1 | Seeds & Oils |
| Melonseed | 299 | 1 | 0 | Seeds & Oils |
| Tallowtree seed | 305 | 1 | 0 | Seeds & Oils |
| Kapok fruit | 310 | 1 | 0 | Seeds & Oils |
| Kapokseed in shell | 311 | 1 | 1 | Seeds & Oils |
| Seed cotton | 328 | 1 | 0 | Seeds & Oils |
| Cottonseed | 329 | 1 | 1 | Seeds & Oils |
| Cake of Cottonseed | 332 | 0 | 1 | Seeds & Oils |

| | | | | |
|---|---|---|---|---|
| Linseed | 333 | 1 | 1 | Seeds & Oils |
| Linseed oil | 334 | 0 | 1 | Seeds & Oils |
| Cake of Linseed | 335 | 0 | 1 | Seeds & Oils |
| Hempseed | 336 | 1 | 0 | Seeds & Oils |
| Oilseeds, Nes | 339 | 1 | 1 | Seeds & Oils |
| Cabbages and other brassicas | 358 | 1 | 1 | Vegetables |
| Artichokes | 366 | 1 | 1 | Vegetables |
| Asparagus | 367 | 1 | 1 | Vegetables |
| Lettuce and chicory | 372 | 1 | 1 | Vegetables |
| Spinach | 373 | 1 | 1 | Vegetables |
| Cassava leaves | 378 | 1 | 0 | Vegetables |
| Tomatoes | 388 | 1 | 1 | Vegetables |
| Juice of Tomatoes | 390 | 0 | 1 | Vegetables |
| Paste of Tomatoes | 391 | 0 | 1 | Vegetables |
| Tomato Peeled | 392 | 0 | 1 | Vegetables |
| Cauliflowers and broccoli | 393 | 1 | 1 | Vegetables |
| Pumpkins, squash and gourds | 394 | 1 | 1 | Vegetables |
| Cucumbers and gherkins | 397 | 1 | 1 | Vegetables |
| Eggplant-baseds (aubergines) | 399 | 1 | 1 | Vegetables |
| Chillies and peppers, green | 401 | 1 | 1 | Vegetables |
| Onions (inc. shallots), green | 402 | 1 | 1 | Vegetables |
| Onions, dry | 403 | 1 | 1 | Vegetables |
| Garlic | 406 | 1 | 1 | Vegetables |
| Leeks, other alliaceous vegetables | 407 | 1 | 1 | Vegetables |
| Beans, green | 414 | 1 | 1 | Vegetables |
| Peas, green | 417 | 1 | 1 | Vegetables |
| Vegetables, leguminous nes | 420 | 1 | 0 | Vegetables |
| String beans | 423 | 1 | 0 | Vegetables |
| Carrots and turnips | 426 | 1 | 1 | Vegetables |
| Okra | 430 | 1 | 0 | Vegetables |
| Maize, green | 446 | 1 | 1 | Vegetables |
| Sweet Corn Frozen | 447 | 0 | 1 | Vegetables |
| Mushrooms and truffles | 449 | 1 | 1 | Vegetables |
| Chicory roots | 459 | 1 | 0 | Vegetables |
| Veg.Prod.Fresh Or Dried | 460 | 0 | 1 | Vegetables |
| Carobs | 461 | 1 | 0 | Vegetables |
| Vegetables fresh nes | 463 | 1 | 1 | Vegetables |
| Vegetables, dried nes | 464 | 0 | 1 | Vegetables |
| Vegetables Preserved Nes | 472 | 0 | 1 | Vegetables |
| Vegetable Frozen | 473 | 0 | 1 | Vegetables |
| Bananas | 486 | 1 | 1 | Fruits |
| Plantains | 489 | 1 | 1 | Fruits |
| Oranges | 490 | 1 | 1 | Fruits |
| Orange juice, single strength | 491 | 0 | 1 | Fruits |
| Tangerines, mandarins, clem. | 495 | 1 | 1 | Fruits |
| Lemons and limes | 497 | 1 | 1 | Fruits |
| Grapefruit (inc. pomelos) | 507 | 1 | 1 | Fruits |
| Juice of Grapefruit | 509 | 0 | 1 | Fruits |
| Citrus fruit, nes | 512 | 1 | 0 | Fruits |
| Citrus juice, single strength | 513 | 0 | 1 | Fruits |
| Apples | 515 | 1 | 1 | Fruits |
| Pears | 521 | 1 | 1 | Fruits |
| Quinces | 523 | 1 | 1 | Fruits |

| | | | | |
|---|---|---|---|---|
| Apricots | 526 | 1 | 1 | Fruits |
| Dry Apricots | 527 | 0 | 1 | Fruits |
| Sour cherries | 530 | 1 | 1 | Fruits |
| Cherries | 531 | 1 | 1 | Fruits |
| Peaches and nectarines | 534 | 1 | 1 | Fruits |
| Plums and sloes | 536 | 1 | 1 | Fruits |
| Plums Dried (Prunes) | 537 | 0 | 1 | Fruits |
| Stone fruit, nes | 541 | 1 | 0 | Fruits |
| Fruit, pome nes | 542 | 1 | 0 | Fruits |
| Strawberries | 544 | 1 | 1 | Fruits |
| Raspberries | 547 | 1 | 0 | Fruits |
| Gooseberries | 549 | 1 | 1 | Fruits |
| Currants | 550 | 1 | 1 | Fruits |
| Blueberries | 552 | 1 | 1 | Fruits |
| Cranberries | 554 | 1 | 1 | Fruits |
| Berries Nes | 558 | 1 | 0 | Fruits |
| Grapes | 560 | 1 | 1 | Fruits |
| Raisins | 561 | 0 | 1 | Fruits |
| Grape Juice | 562 | 0 | 1 | Fruits |
| Wine | 564 | 0 | 1 | Lux-foods |
| Vermouths and Similar | 565 | 0 | 1 | Lux-foods |
| Watermelons | 567 | 1 | 1 | Fruits |
| Other melons (inc.cantaloupes) | 568 | 1 | 1 | Fruits |
| Figs | 569 | 1 | 1 | Fruits |
| Mangoes, mangosteens, guavas | 571 | 1 | 1 | Fruits |
| Avocados | 572 | 1 | 1 | Fruits |
| Pineapples | 574 | 1 | 1 | Fruits |
| Juice of Pineapples | 576 | 0 | 1 | Fruits |
| Dates | 577 | 1 | 1 | Fruits |
| Persimmons | 587 | 1 | 1 | Fruits |
| Cashew apple | 591 | 1 | 1 | Fruits |
| Kiwi fruit | 592 | 1 | 1 | Fruits |
| Papayas | 600 | 1 | 1 | Fruits |
| Fruit, tropical fresh nes | 603 | 1 | 1 | Fruits |
| Fruit Fresh Nes | 619 | 1 | 1 | Fruits |
| Fruit, dried nes | 620 | 0 | 1 | Fruits |
| Fruit Juice Nes | 622 | 0 | 1 | Fruits |
| Coffee, green | 656 | 1 | 1 | Lux-foods |
| Coffee Roasted | 657 | 0 | 1 | Lux-foods |
| Cocoa beans | 661 | 1 | 1 | Lux-foods |
| Cocoa Paste | 662 | 0 | 1 | Lux-foods |
| Cocoa Butter | 664 | 0 | 1 | Lux-foods |
| Cocoapowder and Cake | 665 | 0 | 1 | Lux-foods |
| Chocolate Prsnes | 666 | 0 | 1 | Lux-foods |
| Tea | 667 | 1 | 1 | Lux-foods |
| Maté | 671 | 1 | 1 | Lux-foods |
| Hops | 677 | 1 | 1 | Lux-foods |
| Pepper (Piper spp.) | 687 | 1 | 1 | Lux-foods |
| Chillies and peppers, dry | 689 | 1 | 1 | Lux-foods |
| Vanilla | 692 | 1 | 1 | Lux-foods |
| Cinnamon (canella) | 693 | 1 | 1 | Lux-foods |
| Cloves | 698 | 1 | 1 | Lux-foods |
| Nutmeg, mace and cardamoms | 702 | 1 | 1 | Lux-foods |

| | | | | |
|---|---|---|---|---|
| Anise, badian, fennel, corian. | 711 | 1 | 1 | Lux-foods |
| Ginger | 720 | 1 | 1 | Lux-foods |
| Spices, nes | 723 | 1 | 1 | Lux-foods |
| Peppermint | 748 | 1 | 1 | Lux-foods |
| Cotton lint | 767 | 1 | 1 | Non edible |
| Cotton Carded,Combed | 768 | 0 | 1 | Non edible |
| Cotton Waste | 769 | 0 | 1 | Non edible |
| Cotton Linter | 770 | 0 | 1 | Non edible |
| Flax fibre and tow | 773 | 1 | 1 | Non edible |
| Flax Tow Waste | 774 | 0 | 1 | Non edible |
| Hemp Tow Waste | 777 | 1 | 0 | Non edible |
| Kapok fibre | 778 | 1 | 1 | Non edible |
| Jute | 780 | 1 | 1 | Non edible |
| Other Bastfibres | 782 | 1 | 0 | Non edible |
| Ramie | 788 | 1 | 0 | Non edible |
| Sisal | 789 | 1 | 0 | Non edible |
| Agave Fibres Nes | 800 | 1 | 0 | Non edible |
| Manila Fibre (Abaca) | 809 | 1 | 1 | Non edible |
| Coir | 813 | 1 | 0 | Non edible |
| Fibre Crops Nes | 821 | 1 | 0 | Non edible |
| Tobacco, unmanufactured | 826 | 1 | 1 | Non edible |
| Natural rubber | 836 | 1 | 1 | Non edible |
| Gums, natural | 839 | 1 | 0 | Non edible |
| Cattle | 866 | 0 | 1 | Live animals |
| Cattle meat | 867 | 1 | 1 | Meat |
| Offals of cattle, edible | 868 | 0 | 1 | Meat |
| Fat, cattle | 869 | 0 | 1 | Meat |
| Meat-Cattle, boneless | 870 | 0 | 1 | Meat |
| Sausage Beef and Veal | 874 | 0 | 1 | Meat |
| Meat, beef, preparations | 875 | 0 | 1 | Meat |
| Cow milk, whole, fresh | 882 | 1 | 1 | Diary & Eggs |
| Cream fresh | 885 | 0 | 1 | Diary & Eggs |
| Butter Cow Milk | 886 | 0 | 1 | Diary & Eggs |
| Milk Skm of Cows | 888 | 0 | 1 | Diary & Eggs |
| Milk Whole Cond | 889 | 0 | 1 | Diary & Eggs |
| Whey Condensed | 890 | 0 | 1 | Diary & Eggs |
| Yoghurt, concentrated or not | 892 | 0 | 1 | Diary & Eggs |
| Butterm.,Curdl,Acid.Milk | 893 | 0 | 1 | Diary & Eggs |
| Milk, whole evaporated | 894 | 0 | 1 | Diary & Eggs |
| Milk Whole Dried | 897 | 0 | 1 | Diary & Eggs |
| Milk Skimmed Dry | 898 | 0 | 1 | Diary & Eggs |
| Whey, dry | 900 | 0 | 1 | Diary & Eggs |
| Cheese of Whole Cow Milk | 901 | 0 | 1 | Diary & Eggs |
| Processed Cheese | 907 | 0 | 1 | Diary & Eggs |
| Prod.of Nat.Milk Constit | 909 | 0 | 1 | Diary & Eggs |
| Ice cream and edible ice | 910 | 0 | 1 | Diary & Eggs |
| Meat indigenous, cattle | 944 | 1 | 0 | Meat |
| Buffaloes | 946 | 0 | 1 | Live animals |
| Meat, buffalo | 947 | 1 | 0 | Meat |
| Milk, whole fresh buffalo | 951 | 1 | 0 | Diary & Eggs |
| Ghee, of buffalo milk | 953 | 0 | 1 | Diary & Eggs |
| Meat indigenous, buffalo | 972 | 1 | 0 | Meat |
| Sheep | 976 | 0 | 1 | Live animals |

| | | | | |
|---|---|---|---|---|
| Sheep meat | 977 | 1 | 1 | Meat |
| Offals of Sheep,Edible | 978 | 0 | 1 | Meat |
| Milk, whole fresh sheep | 982 | 1 | 1 | Diary & Eggs |
| Cheese of Sheep Milk | 984 | 0 | 1 | Diary & Eggs |
| Meat indigenous, sheep | 1012 | 1 | 0 | Meat |
| Goats | 1016 | 0 | 1 | Live animals |
| Goat meat | 1017 | 1 | 1 | Meat |
| Offals of Goats, Edible | 1018 | 0 | 1 | Meat |
| Milk, whole fresh goat | 1020 | 1 | 0 | Diary & Eggs |
| Meat indigenous, goat | 1032 | 1 | 0 | Meat |
| Pigs | 1034 | 0 | 1 | Live animals |
| Pig meat | 1035 | 1 | 1 | Meat |
| Offals of Pigs, Edible | 1036 | 0 | 1 | Meat |
| Fat of Pigs | 1037 | 0 | 1 | Meat |
| Meat, pork | 1038 | 0 | 1 | Meat |
| Bacon and Ham | 1039 | 0 | 1 | Meat |
| Sausages of Pig Meat | 1041 | 0 | 1 | Meat |
| Prep of Pig Meat | 1042 | 0 | 1 | Meat |
| Lard | 1043 | 0 | 1 | Meat |
| Meat indigenous, pig | 1055 | 1 | 0 | Meat |
| Chickens | 1057 | 0 | 1 | Live animals |
| Meat, chicken | 1058 | 1 | 1 | Meat |
| Offals, liver chicken | 1059 | 0 | 1 | Meat |
| Fat, liver prepared (foie gras) | 1060 | 0 | 1 | Meat |
| Meat, chicken, canned | 1061 | 0 | 1 | Meat |
| Hen eggs, in shell | 1062 | 1 | 1 | Diary & Eggs |
| Eggs Liquid | 1063 | 0 | 1 | Diary & Eggs |
| Eggs Dried | 1064 | 0 | 1 | Diary & Eggs |
| Ducks | 1068 | 0 | 1 | Live animals |
| Duck meat | 1069 | 1 | 1 | Meat |
| Meat indigenous, duck | 1070 | 1 | 0 | Meat |
| Goose and guinea fowl meat | 1073 | 1 | 1 | Meat |
| Offals, liver geese | 1074 | 0 | 1 | Meat |
| Offals, liver duck | 1075 | 0 | 1 | Meat |
| Meat indigenous, geese | 1077 | 1 | 0 | Meat |
| Turkeys | 1079 | 0 | 1 | Live animals |
| Turkey meat | 1080 | 1 | 1 | Meat |
| Meat indigenous, bird nes | 1084 | 1 | 0 | Meat |
| Meat indigenous, turkey | 1087 | 1 | 0 | Meat |
| Meat, bird nes | 1089 | 1 | 0 | Meat |
| Other bird eggs,in shell | 1091 | 1 | 1 | Diary & Eggs |
| Meat indigenous, chicken | 1094 | 1 | 0 | Meat |
| Horses | 1096 | 0 | 1 | Live animals |
| Horse meat | 1097 | 1 | 1 | Meat |
| Asses | 1107 | 0 | 1 | Live animals |
| Meat, ass | 1108 | 1 | 1 | Meat |
| Mules | 1110 | 0 | 1 | Live animals |
| Meat, mule | 1111 | 1 | 0 | Meat |
| Meat indigenous, horse | 1120 | 1 | 0 | Meat |
| Meat indigenous, ass | 1122 | 1 | 0 | Meat |
| Meat indigenous, mule | 1124 | 1 | 0 | Meat |
| Milk, whole fresh camel | 1130 | 1 | 0 | Diary & Eggs |
| Meat, game | 1163 | 1 | 1 | Meat |

| | | | | |
|---|---|---|---|---|
| Meat, dried nes | 1164 | 0 | 1 | Meat |
| Meat, nes | 1166 | 1 | 1 | Meat |
| Offals, nes | 1167 | 1 | 0 | Meat |
| Oils, fats of animal nes | 1168 | 0 | 1 | Meat |
| Meal, meat | 1173 | 0 | 1 | Meat |
| Tallow | 1225 | 0 | 1 | Meat |

650

### Table A.2  List of countries in the CWASI database

| Country name | FAO code | position | reporting | discontinuities |
|---|---|---|---|---|
| 'Afghanistan' | 2 | 1 | 1 | |
| 'Albania' | 3 | 2 | 1 | |
| 'Algeria' | 4 | 3 | 1 | |
| 'American Samoa' | 5 | 4 | 0 | |
| 'Andorra' | 6 | 5 | 0 | |
| 'Angola' | 7 | 6 | 0 | |
| 'Anguilla' | 258 | 7 | 0 | |
| 'Antarctica' | 30 | 8 | 0 | |
| 'Antigua and Barbuda' | 8 | 9 | 1 | |
| 'Argentina' | 9 | 10 | 1 | |
| 'Armenia' | 1 | 11 | 1 | active from 1992 |
| 'Aruba' | 22 | 12 | 1 | |
| 'Australia' | 10 | 13 | 1 | |
| 'Austria' | 11 | 14 | 1 | |
| 'Azerbaijan' | 52 | 15 | 1 | active from 1992 |
| 'Bahamas' | 12 | 16 | 1 | |
| 'Bahrain' | 13 | 17 | 1 | |
| 'Bangladesh' | 16 | 18 | 1 | |
| 'Barbados' | 14 | 19 | 1 | |
| 'Belarus' | 57 | 20 | 1 | active from 1992 |
| 'Belgium' | 255 | 21 | 1 | active from 2000 |
| 'Belgium-Luxembourg' | 15 | 22 | 1 | inactive from 2000 |
| 'Belize' | 23 | 23 | 1 | |
| 'Benin' | 53 | 24 | 1 | |
| 'Bermuda' | 17 | 25 | 1 | |
| 'Bhutan' | 18 | 26 | 1 | |
| 'Bolivia, Plurinational State of' | 19 | 27 | 1 | |
| 'Bosnia and Herzegovina' | 80 | 28 | 1 | active from 1992 |
| 'Botswana' | 20 | 29 | 1 | |
| 'Bouvet Island' | 31 | 30 | 0 | |
| 'Brazil' | 21 | 31 | 1 | |
| 'British Indian Ocean Territory' | 24 | 32 | 0 | |
| 'British Virgin Islands' | 239 | 33 | 0 | |
| 'Brunei Darussalam' | 26 | 34 | 1 | |
| 'Bulgaria' | 27 | 35 | 1 | |
| 'Burkina Faso' | 233 | 36 | 1 | |
| 'Burundi' | 29 | 37 | 1 | |
| 'Cambodia' | 115 | 38 | 1 | |
| 'Cameroon' | 32 | 39 | 1 | |

| | | | | |
|---|---|---|---|---|
| 'Canada' | 33 | 40 | 1 | |
| 'Canton and Enderbury Islands' | 34 | 41 | 0 | |
| 'Cape Verde' | 35 | 42 | 1 | |
| 'Cayman Islands' | 36 | 43 | 0 | |
| 'Central African Republic' | 37 | 44 | 1 | |
| 'Chad' | 39 | 45 | 0 | |
| 'Chile' | 40 | 46 | 1 | |
| 'China, Hong Kong SAR' | 96 | 47 | 1 | |
| 'China, Macao SAR' | 128 | 48 | 1 | |
| 'China, Mainland' | 41 | 49 | 1 | |
| 'China, Taiwan Province of' | 214 | 50 | 1 | |
| 'Christmas Island' | 42 | 51 | 0 | |
| 'Cocos Islands (Keeling)' | 43 | 52 | 0 | |
| 'Colombia' | 44 | 53 | 1 | |
| 'Comoros' | 45 | 54 | 1 | |
| 'Congo' | 46 | 55 | 1 | |
| 'Congo, Democratic Republic of the' | 250 | 56 | 1 | |
| 'Cook Islands' | 47 | 57 | 1 | |
| 'Costa Rica' | 48 | 58 | 1 | |
| 'Cote de Ivoire' | 107 | 59 | 1 | |
| 'Croatia' | 98 | 60 | 1 | active from 1992 |
| 'Cuba' | 49 | 61 | 1 | |
| 'Cyprus' | 50 | 62 | 1 | |
| 'Czech Republic' | 167 | 63 | 1 | active from 1993 |
| 'Czechoslovakia' | 51 | 64 | 1 | inactive from 1993 |
| 'Denmark' | 54 | 65 | 1 | |
| 'Djibouti' | 72 | 66 | 1 | |
| 'Dominica' | 55 | 67 | 1 | |
| 'Dominican Republic' | 56 | 68 | 1 | |
| 'Ecuador' | 58 | 69 | 1 | |
| 'Egypt' | 59 | 70 | 1 | |
| 'El Salvador' | 60 | 71 | 1 | |
| 'Equatorial Guinea' | 61 | 72 | 0 | |
| 'Eritrea' | 178 | 73 | 0 | active from 1993 |
| 'Estonia' | 63 | 74 | 1 | active from 1992 |
| 'Ethiopia' | 238 | 75 | 1 | active from 1993 |
| 'Falkland Islands (Malvinas)' | 65 | 76 | 0 | |
| 'Faroe Islands' | 64 | 77 | 1 | |
| 'Fiji' | 66 | 78 | 1 | |
| 'Finland' | 67 | 79 | 1 | |
| 'France' | 68 | 80 | 1 | |
| 'French Guiana' | 69 | 81 | 1 | |
| 'French Polynesia' | 70 | 82 | 1 | |
| 'French Southern and Antarctic Territories' | 71 | 83 | 0 | |
| 'Gabon' | 74 | 84 | 1 | |
| 'Gambia' | 75 | 85 | 1 | |
| 'Georgia' | 73 | 86 | 1 | active from 1992 |
| 'Germany' | 79 | 87 | 1 | |
| 'Ethiopia PDR' | 62 | 88 | 1 | inactive from 1993 |
| 'Neutral Zone' | 152 | 89 | 0 | all zeros |
| 'Ghana' | 81 | 90 | 1 | |
| 'Gibraltar' | 82 | 91 | 0 | |
| 'Greece' | 84 | 92 | 1 | |

| | | | | |
|---|---|---|---|---|
| 'Greenland' | 85 | 93 | 1 | |
| 'Grenada' | 86 | 94 | 1 | |
| 'Guadeloupe' | 87 | 95 | 1 | |
| 'Guam' | 88 | 96 | 0 | |
| 'Guatemala' | 89 | 97 | 1 | |
| 'Guinea' | 90 | 98 | 1 | |
| 'Guinea-Bissau' | 175 | 99 | 0 | |
| 'Guyana' | 91 | 100 | 1 | |
| 'Haiti' | 93 | 101 | 0 | |
| 'Heard and McDonald Islands' | 92 | 102 | 0 | |
| 'Holy See' | 94 | 103 | 0 | |
| 'Honduras' | 95 | 104 | 1 | |
| 'Hungary' | 97 | 105 | 1 | |
| 'Iceland' | 99 | 106 | 1 | |
| 'India' | 100 | 107 | 1 | |
| 'Indonesia' | 101 | 108 | 1 | |
| 'Iran, Islamic Republic of' | 102 | 109 | 1 | |
| 'Iraq' | 103 | 110 | 0 | |
| 'Ireland' | 104 | 111 | 1 | |
| 'Israel' | 105 | 112 | 1 | |
| 'Italy' | 106 | 113 | 1 | |
| 'Jamaica' | 109 | 114 | 1 | |
| 'Japan' | 110 | 115 | 1 | |
| 'Johnston Island' | 111 | 116 | 0 | |
| 'Jordan' | 112 | 117 | 1 | |
| 'Kazakhstan' | 108 | 118 | 1 | active from 1992 |
| 'Kenya' | 114 | 119 | 1 | |
| 'Kiribati' | 83 | 120 | 1 | |
| 'Korea, Democratic Peoples Republic of' | 116 | 121 | 0 | |
| 'Korea, Republic of' | 117 | 122 | 1 | |
| 'Kuwait' | 118 | 123 | 1 | |
| 'Kyrgyzstan' | 113 | 124 | 1 | active from 1992 |
| 'Lao Peoples Democratic Republic' | 120 | 125 | 0 | |
| 'Latvia' | 119 | 126 | 1 | active from 1992 |
| 'Lebanon' | 121 | 127 | 1 | |
| 'Lesotho' | 122 | 128 | 0 | |
| 'Liberia' | 123 | 129 | 0 | |
| 'Libya' | 124 | 130 | 1 | |
| 'Liechtenstein' | 125 | 131 | 0 | |
| 'Lithuania' | 126 | 132 | 1 | active from 1992 |
| 'Luxembourg' | 256 | 133 | 1 | active from 2000 |
| 'Macedonia, The former Yugoslav Republic of' | 154 | 134 | 1 | active from 1992 |
| 'Madagascar' | 129 | 135 | 1 | |
| 'Malawi' | 130 | 136 | 1 | |
| 'Malaysia' | 131 | 137 | 1 | |
| 'Maldives' | 132 | 138 | 1 | |
| 'Mali' | 133 | 139 | 1 | |
| 'Malta' | 134 | 140 | 1 | |
| 'Marshall Islands' | 127 | 141 | 0 | |
| 'Martinique' | 135 | 142 | 1 | |
| 'Mauritania' | 136 | 143 | 1 | |
| 'Mauritius' | 137 | 144 | 1 | |
| 'Mayotte' | 270 | 145 | 0 | |

| | | | | |
|---|---|---|---|---|
| 'Mexico' | 138 | 146 | 1 | |
| 'Micronesia, Federated States of' | 145 | 147 | 0 | |
| 'Midway Island' | 139 | 148 | 0 | |
| 'Moldova, Republic of' | 146 | 149 | 1 | active from 1992 |
| 'Monaco' | 140 | 150 | 0 | |
| 'Mongolia' | 141 | 151 | 1 | |
| 'Montenegro' | 273 | 152 | 1 | active from 2006 |
| 'Montserrat' | 142 | 153 | 1 | |
| 'Morocco' | 143 | 154 | 1 | |
| 'Mozambique' | 144 | 155 | 0 | |
| 'Myanmar' | 28 | 156 | 0 | |
| 'Namibia' | 147 | 157 | 1 | |
| 'Nauru' | 148 | 158 | 0 | |
| 'Nepal' | 149 | 159 | 1 | |
| 'Netherlands' | 150 | 160 | 1 | |
| 'Netherlands Antilles' | 151 | 161 | 1 | |
| 'New Caledonia' | 153 | 162 | 1 | |
| 'New Zealand' | 156 | 163 | 1 | |
| 'Nicaragua' | 157 | 164 | 1 | |
| 'Niger' | 158 | 165 | 1 | |
| 'Nigeria' | 159 | 166 | 1 | |
| 'Niue' | 160 | 167 | 0 | |
| 'Norfolk Island' | 161 | 168 | 0 | |
| 'Northern Mariana Islands' | 163 | 169 | 0 | |
| 'Norway' | 162 | 170 | 1 | |
| 'Occupied Palestinian Territory' | 299 | 171 | 0 | |
| 'Oman' | 221 | 172 | 1 | |
| 'Pacific Islands Trust Territory' | 164 | 173 | 0 | |
| 'Pakistan' | 165 | 174 | 1 | |
| 'Palau' | 180 | 175 | 0 | |
| 'Panama' | 166 | 176 | 1 | |
| 'Papua New Guinea' | 168 | 177 | 1 | |
| 'Paraguay' | 169 | 178 | 1 | |
| 'Peru' | 170 | 179 | 1 | |
| 'Philippines' | 171 | 180 | 1 | |
| 'Pitcairn Islands' | 172 | 181 | 0 | |
| 'Poland' | 173 | 182 | 1 | |
| 'Portugal' | 174 | 183 | 1 | |
| 'Puerto Rico' | 177 | 184 | 0 | |
| 'Qatar' | 179 | 185 | 1 | |
| 'Reunion' | 182 | 186 | 1 | |
| 'Romania' | 183 | 187 | 1 | |
| 'Russian Federation' | 185 | 188 | 1 | active from 1992 |
| 'Rwanda' | 184 | 189 | 1 | |
| 'Saint Helena, Ascension and Tristan da Cunha' | 187 | 190 | 0 | |
| 'Saint Kitts and Nevis' | 188 | 191 | 1 | |
| 'Saint Lucia' | 189 | 192 | 1 | |
| 'Saint Pierre and Miquelon' | 190 | 193 | 0 | |
| 'Saint Vincent and the Grenadines' | 191 | 194 | 1 | |
| 'Samoa' | 244 | 195 | 0 | |
| 'San Marino' | 192 | 196 | 0 | |
| 'Sao Tome and Principe' | 193 | 197 | 1 | |
| 'Saudi Arabia' | 194 | 198 | 1 | |

| | | | | |
|---|---|---|---|---|
| 'Senegal' | 195 | 199 | 1 | |
| 'Serbia' | 272 | 200 | 1 | active from 2006 |
| 'Serbia and Montenegro' | 186 | 201 | 1 | active from 1992 to 2005 |
| 'Seychelles' | 196 | 202 | 1 | |
| 'Sierra Leone' | 197 | 203 | 1 | |
| 'Singapore' | 200 | 204 | 1 | |
| 'Slovakia' | 199 | 205 | 1 | active from 1993 |
| 'Slovenia' | 198 | 206 | 1 | active from 1992 |
| 'Solomon Islands' | 25 | 207 | 1 | |
| 'Somalia' | 201 | 208 | 0 | |
| 'South Africa' | 202 | 209 | 1 | |
| 'South Georgia and the South Sandwich Islands' | 271 | 210 | 0 | |
| 'Spain' | 203 | 211 | 1 | |
| 'Sri Lanka' | 38 | 212 | 1 | |
| 'Sudan (former)' | 206 | 213 | 1 | inactive from 2012 |
| 'Suriname' | 207 | 214 | 1 | |
| 'Svalbard and Jan Mayen Islands' | 260 | 215 | 0 | |
| 'Swaziland' | 209 | 216 | 1 | |
| 'Sweden' | 210 | 217 | 1 | |
| 'Switzerland' | 211 | 218 | 1 | |
| 'Syrian Arab Republic' | 212 | 219 | 1 | |
| 'Tajikistan' | 208 | 220 | 0 | active from 1992 |
| 'Tanzania, United Republic of' | 215 | 221 | 1 | |
| 'Thailand' | 216 | 222 | 1 | |
| 'Timor-Leste' | 176 | 223 | 0 | |
| 'Togo' | 217 | 224 | 1 | |
| 'Tokelau' | 218 | 225 | 0 | |
| 'Tonga' | 219 | 226 | 1 | |
| 'Trinidad and Tobago' | 220 | 227 | 1 | |
| 'Tunisia' | 222 | 228 | 1 | |
| 'Turkey' | 223 | 229 | 1 | |
| 'Turkmenistan' | 213 | 230 | 0 | active from 1992 |
| 'Turks and Caicos Islands' | 224 | 231 | 0 | |
| 'Tuvalu' | 227 | 232 | 1 | |
| 'Uganda' | 226 | 233 | 1 | |
| 'Ukraine' | 230 | 234 | 1 | active from 1992 |
| 'United Arab Emirates' | 225 | 235 | 1 | |
| 'United Kingdom' | 229 | 236 | 1 | |
| 'United States Minor Is.' | 232 | 237 | 0 | |
| 'United States Of America' | 231 | 238 | 1 | |
| 'United States Virgin Islands' | 240 | 239 | 0 | |
| 'Unspecified' | 252 | 240 | 0 | all zeros |
| 'Uruguay' | 234 | 241 | 1 | |
| 'Ussr' | 228 | 242 | 1 | inactive from 1992 |
| 'Uzbekistan' | 235 | 243 | 0 | active from 1992 |
| 'Vanuatu' | 155 | 244 | 1 | |
| 'Venezuela, Bolivarian Republic of' | 236 | 245 | 1 | |
| 'Viet Nam' | 237 | 246 | 0 | |
| 'Wake Island' | 242 | 247 | 0 | |
| 'Wallis and Futuna Islands' | 243 | 248 | 0 | |
| 'Western Sahara' | 205 | 249 | 0 | |
| 'Yemen' | 249 | 250 | 1 | |
| 'Sudan' | 276 | 251 | 0 | active from 2012 |

| | | | | |
|---|---|---|---|---|
| 'South Sudan' | 277 | 252 | 0 | active from 2012 |
| 'Yugoslav SFR' | 248 | 253 | 1 | inactive from 1992 |
| 'Zambia' | 251 | 254 | 1 | |
| 'Zimbabwe' | 181 | 255 | 1 | |
| 'China' | 351 | NaN | 0 | |