# Peer review of "Virtual water trade and water footprint of agricultural goods: the 1961-2016 CWASI database"

_Earth System Science Data, 2020_

## Referee Comment (RC1) · Anonymous Referee #1 · 16 Dec 2020

This study describes a very useful database for research on regional and global water resources in agriculture. The authors combine data of crop production, consumption, and trade from the FAO with the WaterStat by Mekonnen and Hoekstra (2010) to generate a dataset of water consumption in agriculture and virtual water trade that is crop specific, country-specific and runs from 1961 to 2016. The manuscript is clear and well written. The methodology used by the authors is sound and robust. The dataset will facilitate and advance research on the globalization of water. I think this is an important contribution that will draw a lot of attention. I support the publication of this manuscript. I only have very few minor points.

[Figure]

Line 14: please remove the funding source from the abstract. I think the abstract should be about the science questions, how they are addressed and the results. It is not a common practice to include acknowledgements in the abstract. Line 21: "15 years ago" Lines 37-40: poorly phrased Line 38: uWF are not data but "estimates" Line 43: "presented in Hoekstra et al..." Line 46: "available from the literature..." Line 82: "quantified in this study aims..." Line 86: "open access database" Line 90: "extent it to provide annual statistics of..." Line 94: " the results of this analysis constitute the CWASI database

---

## Referee Comment (RC2) · Anonymous Referee #2 · 28 Dec 2020

I admire the hard work by authors to deal with numerous data for the analysis of the long-term temporal change of water consumption for crop production. This sort of work is never done by anyone before. However, I have critical concerns on the values of this work that qualify this manuscript to be published in scientific journals as the original article.

- Estimation of temporal changes of water consumption for crop production Authors adopt a very simple approach to estimate water consumption for crop production in a year from 1960-2016; water consumption for the production of a crop by Mekonnen and Hoekstra 2010 is simply extrapolated by multiplying the ratio of crop yield in a year

to that in the year referred by Mekonnen and Hoekstra. This is too much simplified for the analysis of temporal changes of water consumption for crop production. Water consumption is highly dependent on the climate condition of the production area (e.g. precipitation amount and pattern, temperature, soil conditions, etc.), which can temporally vary in a production area. The approach adopted by authors ignores such crucial aspects of water demand for crop production, which would not make any sense and not give any value of the results. Because it is too far from the real condition. The results are just estimation based on the very much simplified assumption, which can not give any legitimacy as the analysis of the past trend of crop water consumption.

- The value of temporal analysis of water consumption of a crop The analysis of the temporal trend of water consumption for crop production is interesting at some point, but there is no clear implication for the users of this data. If we know the past trend of crop water consumption, who can use these data, and what purpose can they have for better water management? Authors simply state that such temporal analysis of crop water consumption has not yet done anywhere, however why we need to know such a temporal trend and how can we use such information to improve our water management? Authors mention the temporal trend of crop water consumption in the results section, but no deep interpretation and implication is given. The only one clear message is crop water consumption per unit production is mostly improved globally, but this is obvious because agricultural technology improves from the past, which can be imagined without this sort of analysis. This concern is strongly related to the first point I raise.

- Crop water consumption is not water footprint Authors use the term, water footprint (WF), but in fact, they only account water consumption for crop production (excluded trade related processes, for instance). The water footprint is defined by the ISO standard 14046:2015, which is obviously different from what is done here. Authors should be careful to use the terms "water footprint".

There are many other minor points, but at the points above, I can not recommend this

article to be published. The work needs more substantial efforts to plan and implement again.

---

## Author Response (AR1)

**FINAL REVIEW**

EDITOR (David Carlson)
Editor initial Comments to the Author:
Very interesting database, exactly the type of expert data compilation efforts we wanted ESSD to highlight. Some residual issues on acronyms, typos, etc., and I suspect we will ask for a table documenting data sources (with DOI, URL, date of last access, etc.). Those issues will probably also emerge from reviews. Thank you for using ESSD.

**We really thank the Editor for his positive feedback and the possibility to revise the manuscript. All data sources have been carefully indicated in the References. If other arrangements are required, we can rearrange the information as needed.**

REVIEWER 1:
Minor text changes + request to remove the funding source from the abstract.

**We thank the Reviewer for his appreciation and the suggestions for improving the text. All corrections have been implemented in the new version of the manuscript and the founding source has now been excluded from the abstract.**

REVIEWER 2:
I admire the hard work by authors to deal with numerous data… However, I have critical concerns.
- **Estimation of temporal changes of water consumption for crop production**
Authors adopt a very simple approach to estimate water consumption for crop production in a year from 1960-2016; water consumption for the production of a crop by Mekonnen and Hoekstra 2010 is simply extrapolated by multiplying the ratio of crop yield in a year to that in the year referred by Mekonnen and Hoekstra. This is too much simplified for the analysis of temporal changes of water consumption for crop production. Water consumption is highly dependent on the climate condition of the production area (e.g. precipitation amount and pattern, temperature, soil conditions, etc.), which can temporally vary in a production area. The approach adopted by authors ignores such crucial aspects of water demand for crop production, which would not make any sense and not give any value of the results. Because it is too far from the real condition. The results are just estimation based on the very much simplified assumption, which can not give any legitimacy as the analysis of the past trend of crop water consumption.

**We thank the Reviewer for recognizing the hard work on the large amount of data. Of course the issue raised by this Reviewer is a relevant one, and at a first sight one could be surprised by the fact that the temporal variations of the water footprint are dominated by yield variations rather that by the temporal variations of evapotranspiration at the yearly scale (by considering the two factors composing the crop water footprint). This finding has been already substantiated in Tuninetti et al. (2017) via a robust data-based analysis (further details and extensions are found below), but some comments are indeed deserved regarding the reason why this occurs, which are mainly two in our understanding: (i) Evapotranspiration fluctuates more intensely in space than in time, due to its strong relation with the radiation balance; in the method we adopt, spatial variations of evapotranspiration are accounted for by referring to the data by Mekonnen and Hoekstra (2010). (ii) In contrast, crop yield fluctuations are very intense both in space and in time, since crop yield variations adsorb the variability of agronomical techniques, mechanization, soil fertility, and also water availability (among others). By getting back to**

the comment "Water consumption is highly dependent on the climate condition of the production area (e.g. precipitation amount and pattern, temperature, soil conditions, etc.), which can temporally vary in a production area. The approach adopted by authors ignores such crucial aspects of water demand for crop.", we can reassure this Reviewer that indeed our approach does not ignore such crucial aspects. We are sorry for not having clearly outlined these points in the previous version of the manuscript: the text will be amended to better explain our approach. Some additional details on this issue are reported below.

The method used to estimate the temporal evolution of the unit water footprint, or Fast-Track method, is a simplified method introduced in a peer reviewed publication (Tuninetti et al., 2017). There, the Fast-Track method has been verified by comparing the unit water footprint obtained with the simplified approach with estimates obtained by applying a complete water footprint estimation model based on a daily soil water balance fed by year-specific hydro-climatic variables. Despite year to year variations, errors associated to the Fast-Track method, i.e. to the hypothesis of keeping a constant evapotranspiration and let only agricultural yield change, are within a 10% range. This error is comparable to the one affecting estimates based on different models and is therefore negligible in practical terms. We refer to the original paper (Tuninetti et al., 2017) for details and specifications.

The Tuninetti's paper has been the basis for the database presented in this manuscript; it underwent a full peer review process and it is now well regarded in the relevant literature, with none of the citing papers highlighting problems with the method.

For the Reviewer's reference, we are able to include here some additional material about the Fast Track method, further reinforcing our point. We attach below a validation for some additional crops, considering years 1961, 1971, 1981 and all years in the range 1991-2004. The choice of additional crops aims at covering a large fraction of food production worldwide and at diversifying the crops' characteristics (growing seasons, sensibility to water stress and fraction of irrigated production). The differences in unit WF are comparable to the results in Tuninetti et al (2017) and although the variability (error) may occasionally be larger than 10%, the results further confirm the great performance of the Fast Track approach on a wide range of crops.

In addition to the above considerations, another point should be mentioned. We agree with the Reviewer that water consumption of crops (evapotranspiration) is dependent on hydro-climatic conditions (precipitation, temperature, …). However the unit water footprint is less sensitive to them, because it is defined as the ratio between evapotranspiration and agricultural yield, both reacting to hydro-climatic fluctuations with equal signs (see, e.g., Doorenbos et al, 1979). Furthermore, it should be noticed that the sum of green and blue water volumes (the "consumptive" water footprint) is considered in the present database. While the separate contributions of green and blue water may be more affected by year-specific hydro-climatic conditions, the sum of the two terms is less sensitive to these conditions, further reducing the overall error associated to the simplified estimation of $uWF$ with the Fast Track method.

As mentioned, the strengths and weaknesses in the approach used to derive the database have now been stated more clearly in Section 3.1 and in the Conclusions. Also, we have added a cautionary note in the Conclusions to use single-year data with care and to put them in a temporal perspective or multi-year average, in order to avoid misinterpretations of year-specific results.

[Figure]

**Figure.** Country-scale comparison of the annual unit WF estimates obtained by the Fast Track approach, CWF(Y), with the values from the detailed methodology accounting for both yield and ET variations, CWF(Y,ET).

**- The value of temporal analysis of water consumption of a crop**

The analysis of the temporal trend of water consumption for crop production is interesting at some point, but there is no clear implication for the users of this data. If we know the past trend of crop water consumption, who can use these data, and what purpose can they have for better water management? Authors simply state that such temporal analysis of crop water consumption has not yet done anywhere, however why we need to know such a temporal trend and how can we use such information to improve our water management? Authors mention the temporal trend of crop water consumption in the results section, but no deep interpretation and implication is given. The only one clear message is crop water consumption per unit production is mostly improved globally, but this is obvious because agricultural technology improves from the past, which can be imagined without this sort of analysis. This concern is strongly related to the first point I raise.

A large body of literature based on the WaterStat database uses values averaged over a decade centered on year 2000. The time-varying database that we are proposing enables quantitative analyses in different periods, and in particular it enables analyses about more recent years, with updated unit water footprint values. The water footprint and virtual water trade literature includes many contributions that address a temporal evolution without including the time-varying unit WF: these studies can be updated and discussed in view of the new data and additional knowledge that is becoming available to the scientific community.

**Moreover, some literature ("at its infancy", cit. from Hoekstra, 2017) is moving in the direction of making projections and future scenarios of water use for food production and trade. How can we try to predict the future without knowing about the past evolution? The database may therefore serve as a unique starting point for any analyses considering the temporal evolution in the past and in the future of the crop water footprint, with very significant potential implication for the users of the database.**

**- Crop water consumption is not water footprint**
Authors use the term, water footprint (WF), but in fact, they only account water consumption for crop production (excluded trade related processes, for instance). The water footprint is defined by the ISO standard 14046:2015, which is obviously different from what is done here. Authors should be careful to use the terms "water footprint".

**The database includes, for as many products and years as possible, a differentiation between the unit WF of production and the unit WF of consumption. The latter takes into account, besides the locally produced goods, also the imported fraction of goods considering their country of origin, based on the procedure proposed by Kastner et al (2011). In this way, the unit WF takes into account the unit WF of the goods in the countries of production and the role of international trade.**
**As for the use of the "Water Footprint" terms, we are aware of ==the ISO standard, which has now been mentioned in the manuscript== but we are also aware of the large body of literature about the water footprint that originated from prof. Hoekstra's work and the Water Footprint Network. We are following his same approach, taking a bottom-up approach as specified in the Introduction. However, with the important improvements introduced in this manuscript (i.e., the separation between the production and supply unit WF and the tracing back of origin country of traded goods) we are improving upon previous limitations of this approach, bridging towards the top-down approach and offering a new database at the state-of-the-art in the field.**

**CITED REFERENCES**

Doorenbos, J., Kassam, A., Bentvelsen, C., Yield response to water, FAO Irrigation and Drainage Paper, Food and Agriculture Organization, 1979.

Hoekstra, A.: Water footprint assessment: evolvement of a new research field, Water Resour. Manage., 31, 3061-3081, doi: 10.1007/s11269-017-1618-5, 2017.

Kastner, T., Kastner, M., Nonhebel, S.: Tracing distant environmental impacts of agricultural products from a consumer perspective, Ecol. Econ., 70 (6), 1032-1040, doi: 10.1016/j.ecolecon.2011.01.012, 2011.

Tuninetti, M., Tamea, S., Laio, F., and Ridolfi, L.: A Fast Track approach to deal with the temporal dimension of crop water footprint, Environ. Res. Lett., 12 (7), 074010, doi: 10.1088/1748-9326/aa6b09, 2017.

---

## Author Response (AR2)

**RESPONSE TO EDITOR**

EDITOR (David Carlson) - Comments to the Author:
Thank you for good response to reviewer comments. Manuscript needs:
a) A data source table. E.g. from Section 2 on commodities, countries, trade matrices, etc. add a table listing: each source by URL and DOI (if available), full citation details, version numbers, date of last access, etc. You want to make explicitly clear to users what you started from, especially as you intend that the CWASI database evolves via additional contributions. You undoubtedly have all this information but you need to provide it in convenient format for users.
b) Add a section on cautions or limitations. Some of this you now include in Section 5 Results, but - to judge utility for their own work - future users will need to read your cautions. ESSD readers expect a section on uncertainties / cautions / limitations, etc.
c) Formatting errors around line 480. You want to fix these now rather than hoping to catch them later during proofreading.
d) Use the user-friendly version of data DOI, e.g. https://doi.org/10.xxxxx
e) Consider taking long tables now included in supplement as Appendices instead. This change because Copernicus will archive full manuscript but not, in the long-term, supplements.

We are grateful to the Editor for the constructive revision round and for the additional comments which have helped improving the manuscript. Modifications which have now been implemented include:
a) a new data source table (Table 1), detailed by variable and by source web page; References at the end of the manuscript have also been detailed accordingly;
b) a new section on "Uncertainty and limitations", which has now been added after the Results section;
c) the formatting of given paragraph has been fixed;
d) the database DOI has been adjusted in the Abstract and Data Availability sections as required. The DOIs of publications in the reference list have been left in the Copernicus template formatting style;
e) the tables in the supplementary material have now been included in an appendix (Appendix A). The last table (former SI.3) has also been included as text, maintaining all necessary information without repetitions.

Other minor arrangements have been introduced to refine the text or to fix minor issues arisen during proof-reading. The last figure (Figure 8) has also been removed for having been outdated and not much informative.

---

## Author Response (AR3)

**MESSAGE TO THE EDITOR**

Dear Editor,

please accept the submission of final files regarding the accepted manuscript "Virtual water trade and water footprint of agricultural goods: the 1961-2016 CWASI database". The delay was caused by a minimal, but data-intensive adjustment involving two figures motivated by the introduction of a fix in the database (some live animals data were missing) and the exclusion of data that are not part of the published database. The database version on the Zenodo repository has been updated and all details tracked and references updated. We apologize for the delay and we hope not to create further delays to the publication.

Best regards,

Stefania Tamea
* * *
Stefania Tamea, Ph.D.
Associate Professor of Hydrology
Dipartimento di Ingegneria dell'Ambiente, del Territorio e delle Infrastrutture (DIATI)
Politecnico di Torino
Corso Duca degli Abruzzi, 24
10129 Torino
Italy
tel: +39-011-090-5670
fax: +39-011-090-5698
e-mail: stefania.tamea@polito.it
* * *